# KIF2C regulates synaptic plasticity and cognition in mice through dynamic microtubule depolymerization

Rui Zheng[1,2†], Yonglan Du[1,2†], Xintai Wang[2], Tailin Liao[1,2], Zhe Zhang[1,2], Na Wang[1,2], Xiumao Li[3], Ying Shen[4], Lei Shi[5], Jianhong Luo[1,2*], Jun Xia[6*], Ziyi Wang[7*], Junyu Xu[1,2*]

[1]Department of Neurobiology and Department of Rehabilitation of the Children's Hospital, Zhejiang University School of Medicine, Hangzhou, China; [2]NHC and CAMS Key Laboratory of Medical Neurobiology, Ministry of Education Frontier Science Center for Brain Research and Brain Machine Integration, School of Brain Science and Brain Medicine, Zhejiang University, Hangzhou, China; [3]Department of Orthopedic Surgery, the Second Affiliated Hospital, Zhejiang University School of Medicine, Hangzhou, China; [4]Department of Physiology and Department of Neurology of First Affiliated Hospital, Zhejiang University School of Medicine, Hangzhou, China; [5]JNU-HKUST Joint Laboratory for Neuroscience and Innovative Drug Research, Jinan University, Guanzhou, China; [6]Division of Life Science and The Brain and Intelligence Research Institute, The Hong Kong University of Science and Technology, Hong Kong, China; [7]Innovative Institute of Basic Medical Sciences of Zhejiang University (Yuhang), Hangzhou, China

*For correspondence:
luojianhong@zju.edu.cn (JL);
jxia@ust.hk (JX);
mtray@qq.com (ZW);
junyu@zju.edu.cn (JX)

†These authors contributed equally to this work

Competing interest: The authors declare that no competing interests exist.

**Abstract** Dynamic microtubules play a critical role in cell structure and function. In nervous system, microtubules are the major route for cargo protein trafficking and they specially extend into and out of synapses to regulate synaptic development and plasticity. However, the detailed depolymerization mechanism that regulates dynamic microtubules in synapses and dendrites is still unclear. In this study, we find that KIF2C, a dynamic microtubule depolymerization protein without known function in the nervous system, plays a pivotal role in the structural and functional plasticity of synapses and regulates cognitive function in mice. Through its microtubule depolymerization capability, KIF2C regulates microtubule dynamics in dendrites, and regulates microtubule invasion of spines in neurons in a neuronal activity-dependent manner. Using RNAi knockdown and conditional knockout approaches, we showed that KIF2C regulates spine morphology and synaptic membrane expression of AMPA receptors. Moreover, KIF2C deficiency leads to impaired excitatory transmission, long-term potentiation, and altered cognitive behaviors in mice. Collectively, our study explores a novel function of KIF2C in the nervous system and provides an important regulatory mechanism on how activity-dependent microtubule dynamic regulates synaptic plasticity and cognition behaviors.

## Editor's evaluation

In this manuscript, the authors report a set of exciting and novel data suggesting a critical role of the microtubule depolymerizing kinesin-13 KIF2C/MCAK in glutamate receptor trafficking, synaptic plasticity and behaviours related to learning and memory. The authors use multiple experimental approaches, ranging from electron microscopy to mouse behavioral analysis to support the conclusions, which are convincing and provide novel insights into the in vivo

functions of KIF2C in the regulation of excitatory synaptic structure and function and cognitive behaviors.

## Introduction

Microtubules (MTs) are eukaryotic cytoskeletons that play an important role in cell function. In the central nervous system, MTs serve as a structural basis for trafficking material throughout the neuron and are the primary regulators of neuronal differentiation (*Dent and Baas, 2014*). The MTs dynamic instability, that is the growing or shrinking of MT plus end, has been detected in both shaft and dendritic protrusions (*Mitchison and Kirschner, 1984*) and is involved in spine formation and synaptic function (*Hu et al., 2008*; *Gu et al., 2008*; *Merriam et al., 2011*; *Tada and Sheng, 2006*). Inhibition of MT polymerization results in deficits in long-term potentiation (LTP) and negatively affects memory formation or retention (*Shumyatsky et al., 2005*; *Jaworski et al., 2009*). The abnormal MTs dynamics is associated with several neurodegenerative and psychiatric diseases, including Alzheimer's disease, Parkinson's disease, schizophrenia, and depression (*Brunden et al., 2014*). During neuronal activities, MTs can be regulated to extend from the dendritic shaft into spines and affect synaptic structure and function (*3-8*), which could be an important event for synaptic plasticity. However, regulatory mechanism underlying MT dynamics in or out from synapse upon neuronal activity is still unclear, nor its role in learning and memory.

The polymerization dynamics of MTs are essential for their biological functions and can be regulated by many MT-associated proteins (MAPs) (*Akhmanova and Steinmetz, 2015*; *Bowne-Anderson et al., 2015*). The kinesin-13 family of kinesin superfamily proteins (KIFs) has a distinct function in depolymerizing MTs (*Howard and Hyman, 2007*), and therefore is involved in many MT-dependent events (*Kline-Smith et al., 2004*; *Homma et al., 2003*). KIF2C, also called mitotic centromere-associated kinesin (MCAK), is the best-characterized family member and is localized to the spindle poles, spindle midzone, and kinetochores in dividing cells. KIF2C utilizes MT depolymerase activity to regulate spindle formation and correct inappropriate MT attachments at kinetochores (*Gorbsky, 2004*; *Helenius et al., 2006*; *Knowlton et al., 2006*). KIF2C uses ATP hydrolysis for MT depolymerization from both ends and directly interacts with EBs to enhance its destabilizing ability (*Wordeman, 2005*; *Walczak, 2003*; *Wang et al., 2015*; *Moore et al., 2005*; *Lee et al., 2008*). Particularly, KIF2C is associated with Alzheimer's disease and suicidality psychiatric disorders. In vitro study showed that Oligomeric amyloid-β (Aβ) directly inhibits the MT-dependent ATPase activity of KIF2C and leads to abnormal mitotic spindles, which may cause chromosome mis-segregation and increased aneuploid neurons and thereby contributes to the development of Alzheimer's disease (*Borysov et al., 2011*). Clinical evidence shows that KIF2C is significantly decreased in suicidal ideation cohort and could be utilized as a reference for predicting suicidality (*Niculescu et al., 2015*). However, the in vivo physiological function of KIF2C in the CNS remains to be elucidated.

In this study, we have found that KIF2C in the CNS is expressed in a brain-wide and synaptic localized pattern, which could be regulated by neuronal activity. Using a nervous-system-specific KIF2C conditional knockout (cKO) mouse, we further confirmed its pivotal role in spine morphology, synaptic transmission, and synaptic plasticity. By re-expression of KIF2C wild-type (WT) and MT depolymerization-defective mutation, we have found that KIF2C regulates synaptic plasticity by mediating MT depolymerization and affecting MT synaptic invasion. Furthermore, KIF2C cKO mice exhibit deficiencies in multiple cognition-associated tasks, which could be rescued by re-expression of KIF2C(WT) but not the MT depolymerization-defective mutant. Overall, our study explores a novel function of KIF2C in neurons and synaptic plasticity, and reveals how MT dynamics in synapses upon neuronal activity contributes to cognition.

## Results

### KIF2C is highly expressed in the brain and at the synapse

Previous studies have showed that KIF2C plays an important role during cell division in mitotic cells (*Andrews et al., 2004*; *Lan et al., 2004*; *Sanhaji et al., 2014*); however, it is not clear whether KIF2C is also present in the nervous system. Western blot analyses showed that KIF2C was expressed in multiple regions of the brain (*Figure 1A*), and shows sustained high expression in the hippocampus throughout

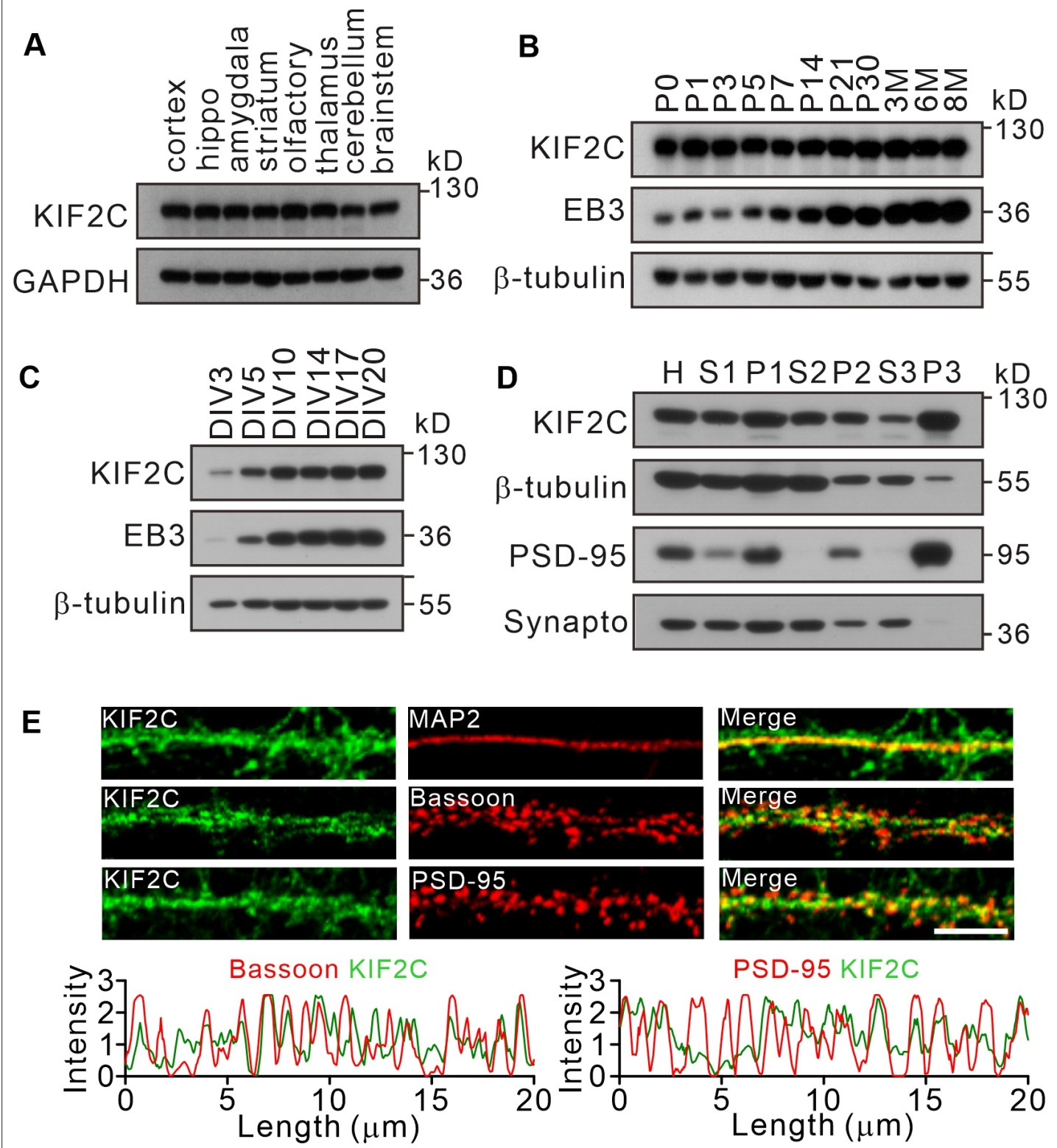

**Figure 1.** Expression of KIF2C in the central nervous system. (**A**) Expression of KIF2C in different mouse brain regions by immunoblotting. (**B**) Expression of KIF2C and EB3 in mouse hippocampus at different day in vitro. (**C**) Developmental expression patterns of KIF2C and EB3 in cultured hippocampal neurons. (**D**) Subcellular distribution of KIF2C in mouse hippocampus. H, homogenate; S1, low-speed supernatant; P1, nuclei; S2, microsomal fraction; P2, synaptosomal fraction; S3, presynaptosomal fraction; P3, postsynaptic density and synapto, synaptophysin. (**E**) Representative images of hippocampal neurons double-labeled with antibodies against KIF2C, PSD-95 (postsynaptic marker) and Bassoon (presynaptic marker). Scale bar, 5 µm.

*Figure 1 continued on next page*

*Figure 1 continued*

The online version of this article includes the following source data for figure 1:

**Source data 1.** Original files of blots with the relevant bands.

the postnatal stages (*Figure 1B*). To identify KIF2C abundance in neurons, western blot analyses were performed on whole-cell extracts of cultured hippocampal neurons (*Figure 1C*). The results showed that KIF2C was expressed in a development-related pattern in cultured neurons: detectable at day in vitro 3 (DIV 3) and significantly increased at DIV 10. Moreover, the subcellular fractionation analysis showed an enrichment of KIF2C in the postsynaptic density (PSD) fraction (*Figure 1D*). Immunofluorescent staining of cultured neurons revealed KIF2C expression as granular puncta of different sizes along dendrites overlapping with the postsynaptic marker PSD-95 and presynaptic protein Bassoon (*Figure 1E*). These results show that KIF2C is abundantly expressed in CNS neurons and is enriched at postsynaptic density, suggesting a possible function in synapses.

## KIF2C knockdown affects spine morphology, synaptic transmission and long-term potentiation in vitro and in vivo

To explore the function of KIF2C in synapses, KIF2C was knocked down in cultured hippocampal neurons using lentiviral-mediated KIF2C shRNA (sh*kif2c*) (*Figure 2A*, *Figure 2—figure supplement 1A*) at DIV 7–10 and the spine morphology was examined at DIV 18–20. The results showed that KIF2C knockdown did not affect spine density (*Figure 2B*). Electrophysiological recordings also showed similar frequency and amplitude of miniature excitatory synaptic currents (mEPSCs) in control and knockdown neurons (*Figure 2—figure supplement 1B*, C), suggesting that acute knockdown of KIF2C in cultured neurons does not affect basal synaptic transmission. We then investigated whether KIF2C regulates spine morphology in an activity-dependent manner. The induction of LTP results in changes in spine structure (*Fifková, 1985*; *Oertner and Matus, 2005*; *Vlachos et al., 2009*; *Fortin et al., 2010*). To test whether KIF2C is involved in spine structural plasticity, we further conducted a glycine-induced chemical LTP (cLTP) assay to produce synaptic enhancements that mimic LTP (*Lu et al., 2001*) in KIF2C knockdown hippocampal neurons. Spine density in knockdown neurons was consistent with control after cLTP stimulation (*Figure 2C and D*). However, the spine head width significantly increased in control hippocampal neurons, whereas it was not affected in the KIF2C knockdown neurons after cLTP stimulation (*Figure 2E*), indicating that KIF2C is essential for spine structural plasticity. In addition, both subcellular fractionation and immunocytochemical results showed that KIF2C was not colocalizing as much with PSD-95 and expressed more in dendritic shafts after cLTP (*Figure 2F–I*), suggesting a translocation of KIF2C from synaptic protrusions into dendrites. Together, these data indicate that KIF2C is removed from postsynaptic sites by neuronal activity, which may contribute to synaptic plasticity.

To further determine the effect of KIF2C on synaptic function, KIF2C was knocked down in hippocampal CA1 neurons in vivo and the electrophysiological properties in acute slices were measured. Mice were injected with either AAV-scramble shRNA (control) or AAV-KIF2C shRNA (sh*kif2c*) in the CA1 region (*Figure 2—figure supplement 1D*), and mEPSCs were recorded. Compared to the control, KIF2C knockdown neurons had enhanced amplitude but similar frequency of mEPSCs (*Figure 3A*). This discrepancy between cultured neurons and brain slices may result from activity-dependent synaptic scaling in free-behaving mice after viral injection. KIF2C knockdown blocked cLTP-induced increased spine width (*Figure 2E*), so we wondered whether the increased mEPSCs amplitude after cLTP stimulation could also be blocked in sh*kif2c* neurons. Hippocampal CA1 slices were incubated with ACSF or glycine (200 µM) for 10 min before mEPSCs recording. The results showed that mEPSCs amplitude was significantly increased in control hippocampal neurons, whereas it was not affected in the sh*kif2c* neurons after cLTP stimulation (*Figure 2—figure supplement 1E, F*). These data indicate that KIF2C knockdown blocks the raise of mEPSCs amplitude induced by cLTP. Furthermore, we investigated whether KIF2C regulated presynaptic function retrogradely by comparing paired-pulse ratios (PPRs). KIF2C knockdown neurons exhibited comparable PPRs to the control, suggesting that KIF2C had a minimal effect on contacting presynaptic function (*Figure 3B*). In addition, the excitatory postsynaptic current ratio of N-methyl-D-aspartic acid receptor (NMDAR)/α-amino-3-hydroxy-5-methyl-4-isoxazolepropionic acid receptor (AMPAR) was decreased in knockdown hippocampal

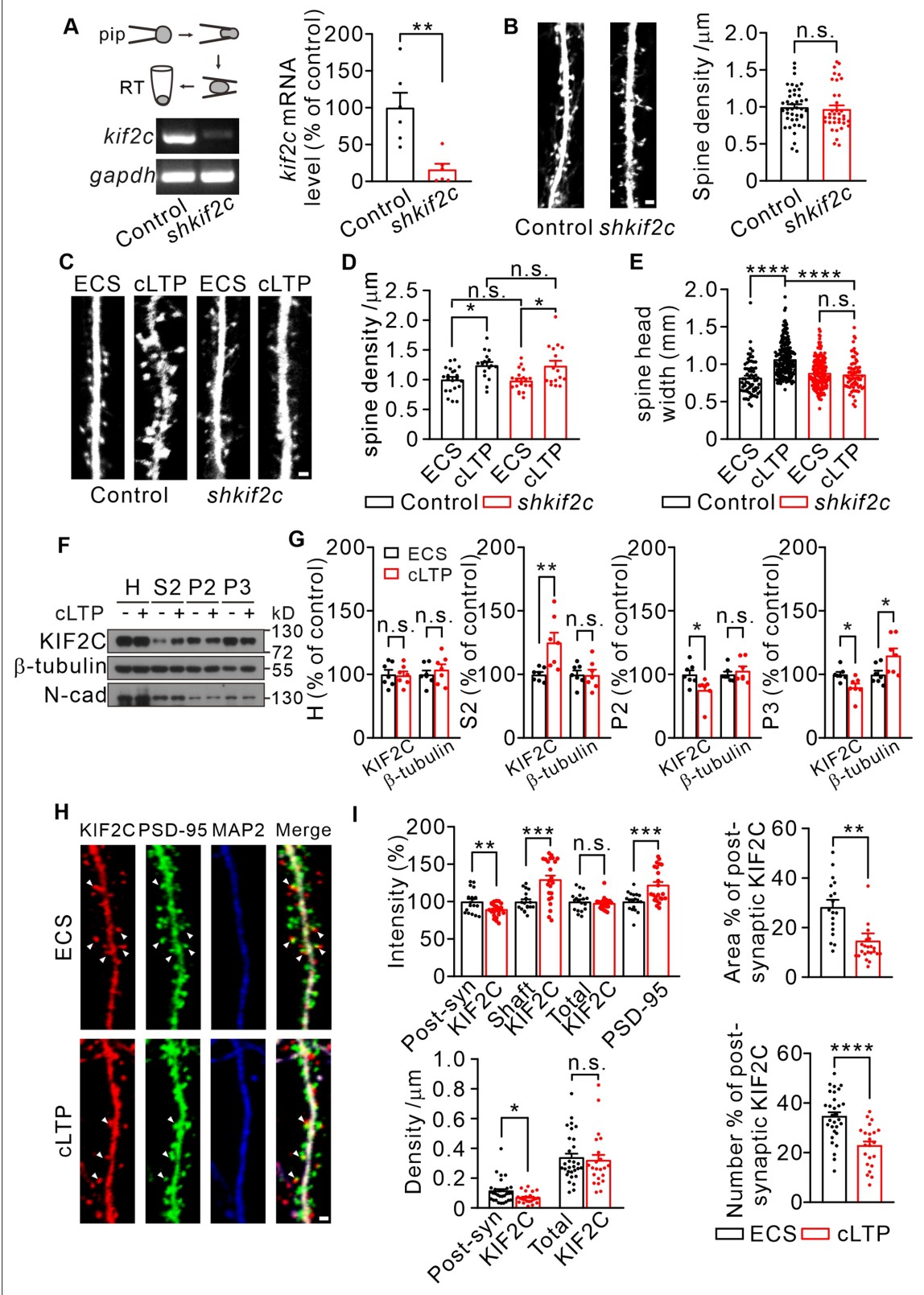

**Figure 2.** KIF2C displays translocation after cLTP and is required for synaptic structure plasticity. (**A**) Electrophoresis of *kif2c* and *gapdh* amplicons from individual control and sh*kif2c* hippocampal neurons. Histograms show percentage changes of *kif2c* mRNA levels (% of control, **p < 0.01, n = 6, Student's *t*-test). (**B**) Cultured hippocampal neurons were infected by control shRNA and sh*kif2c* lentivirus at DIV 7–10, and captured at DIV 18–20. GFP was used to label neuron volume. Scale bar, 1 μm. Spine number per 1 μm (right), n = 20 neurons from three independent culture. Student's *t*-test.

*Figure 2 continued on next page*

*Figure 2 continued*

(**C–E**) Cultured hippocampal neurons were infected by control shRNA and sh*kif2c* lentivirus at DIV 7–10, and incubated with ECS or glycine (200 µM) for 10 min at DIV 18–20. Scale bar, 1 µm. Spine density (**D**) and spine head width (**E**) are calculated. Spine density, n = 17 neurons from three independent culture. Spine head width, n = 24 neurons from three independent culture. n.s., p > 0.05, *p < 0.05, ****p < 0.0001; One way-ANOVA for spine density and head width (*post hoc* comparison).(**F**) Subcellular fraction of KIF2C in cultured hippocampus neurons after 10 min of glycine treatment. N-cadherin (N-cad) was served as negative control. H, Homogenate; S2, microsomal fraction; P2, synaptosomal fraction and P3, postsynaptic density from ECS and cLTP treated neurons. (**G**) Quantification of KIF2C and β-tubulin accumulation with or without cLTP treatment in each fraction. *p < 0.05, **p < 0.01; Student's *t*-test, n = 7 experimental repeats. (**H**) Representative images of co-localization of KIF2C (red), PSD-95 (green), and Microtubule associated protein 2 (MAP2) (blue). Scale bar, 1 µm. (**I**) The percentage of KIF2C puncta that were colocalized with PSD-95 puncta decreased after cLTP treatment. n = 20 neurons from three cultures. n.s., p > 0.05, *p < 0.05, **p < 0.01, ***p < 0.001, ****p < 0.0001; Student's *t*-test.

The online version of this article includes the following source data and figure supplement(s) for figure 2:

**Source data 1.** Values for *kif2c* mRNA levels; Values for spine density and spine head width before or after cLTP stimulation in control and sh*kif2c* neurons; Values for KIF2C synaptic expression before or after cLTP stimulation.

**Source data 2.** Original files of gels and blots with the relevant bands.

**Figure supplement 1.** Miniature excitatory synaptic currents (mEPSCs) recordings of control and sh*kif2c* neurons.

**Figure supplement 1—source data 1.** Values for KIF2C protein levels.

**Figure supplement 1—source data 2.** Original files of blots with the relevant bands.

slices (*Figure 3C*). The input-output relationship of NMDAR/AMPAR-mediated EPSCs showed that amplitude of NMDAR-mediated EPSCs under various stimulation intensities was comparable, while AMPAR-mediated EPSCs was increased in KIF2C knockdown neurons (*Figure 3D–E*), indicating a specific change in AMPAR-mediated synaptic currents. To test whether synaptic plasticity was affected in KIF2C knockdown neurons, we measured whole-cell LTP in CA1 pyramidal neurons by stimulating Schaffer collaterals. The magnitude of LTP was significantly attenuated in knockdown hippocampal slices (*Figure 3F*). These results suggest that KIF2C is necessary for synaptic plasticity in hippocampal neurons, and its insufficiency may affect the basal function of AMPARs in the hippocampus.

## KIF2C knockout affects spine morphology, synaptic transmission, and long-term potentiation in vivo

To further investigate the functions of KIF2C in vivo, we generated cKO mice using the CRISPR/Cas9 system (*Figure 4A*). To obtain cKO mice, *kif2c^flox/flox* mice were mated with *Nestin^Cre* transgenic mice, which resulted in KIF2C deletion only in the nestin cell lineage (*Figure 4—figure supplement 1A*). The knockout efficiency was confirmed using a hippocampal tissue RT-PCR assay (*Figure 4—figure supplement 1B*). There were no significant differences in body weight or brain size between *kif2c^flox/flox;Nestin^Cre* (cKO) and *kif2c^flox/flox* (WT) mice (*Figure 4—figure supplement 1C*). In addition, using Nissl staining, we found that the thickness of hippocampal CA1 and CA3 regions did not change significantly in cKO mice (*Figure 4B*). Further examination of neuronal morphology of CA1 pyramidal cells by Golgi staining and Sholl analysis showed no significant differences in dendritic complexity or spine density between WT mice and KIF2C cKO mice (*Figure 4C–E*). However, the density of mushroom-type spines decreased in basal and apical dendrites of hippocampal neurons in cKO mice, whereas the density of filopodia-type spines increased (*Figure 4F*). Furthermore, electron microscopy analysis revealed that the length and thickness of the PSD fraction were remarkably reduced in the hippocampus of cKO mice (*Figure 4G and H*). Mushroom spines are commonly considered mature spines (*Bourne and Harris, 2008*; *Rochefort and Konnerth, 2012*); therefore, these results indicate impaired spine maturation in the hippocampal CA1 region of cKO mice.

To determine whether such impaired spine maturation may affect synaptic transmission and synaptic plasticity in cKO mice, mEPSCs were first examined in CA1 hippocampal pyramidal neurons. Consistent with the shRNA knockdown experiments in CA1, the amplitude of mEPSCs increased in cKO mice compared to that in WT controls, whereas the frequency of mEPSCs was similar between the two groups (*Figure 5A*). The increased mEPSC amplitude was not due to altered presynaptic activity as the PPRs were not affected in cKO neurons (*Figure 5B*), indicating a postsynaptic influence in mEPSC property. Therefore, we examined the NMDAR/AMPAR ratio as well as the input-output relationship of NMDAR/AMPAR-mediated EPSCs. Result showed a decreased ratio NMDAR/AMPAR in cKO mice (*Figure 5C*). The amplitude of NMDAR-mediated EPSCs was comparable under

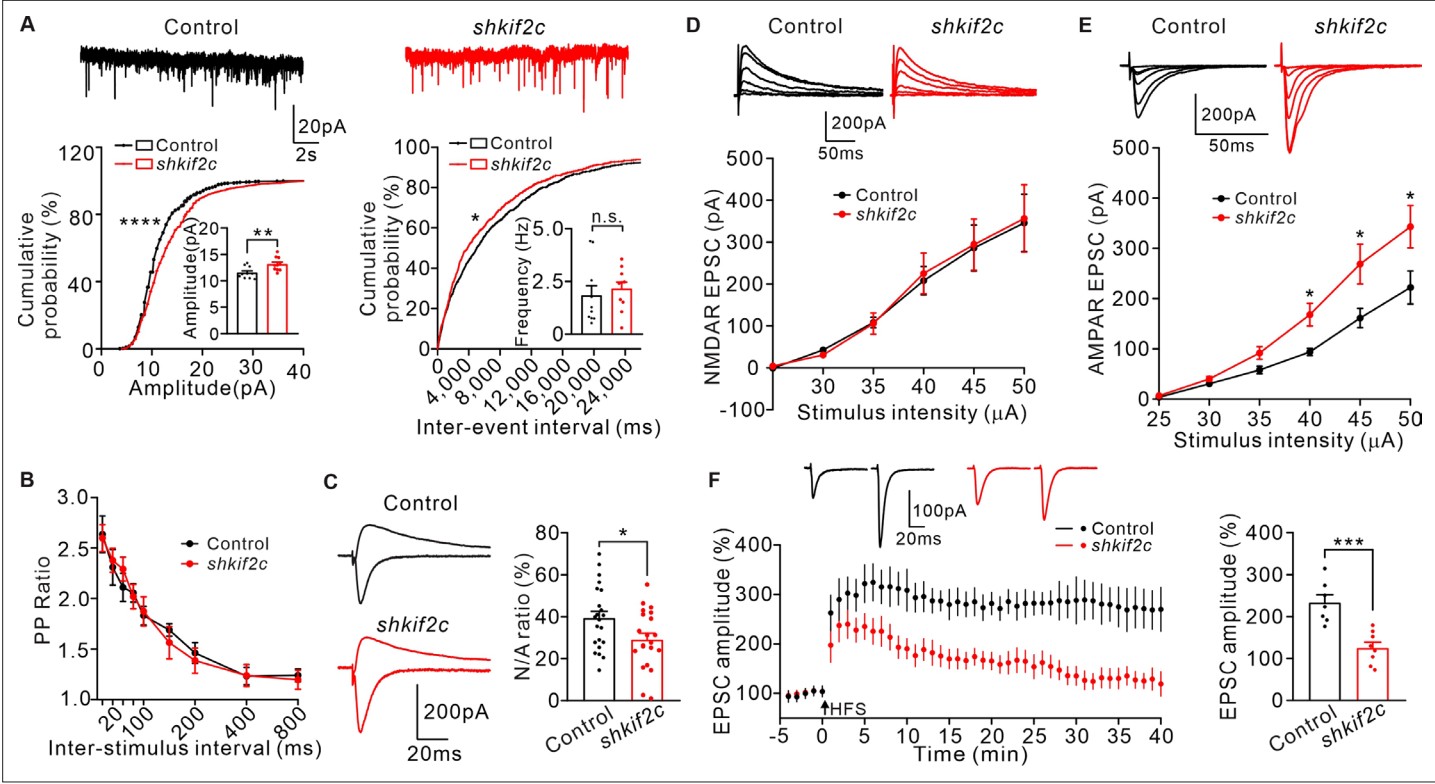

**Figure 3.** Knockdown KIF2C impairs synaptic transmission and plasticity. (**A**) mEPSCs recorded from control and sh*kif2c* CA1 pyramidal neurons. n = 10 per group, n.s., p > 0.05, **p < 0.01, Student's *t*-test; For the cumulative probability curves, *p < 0.05, ****p < 0.0001, Kolmogorov-Smirnov test. (**B**) Paired-pulse stimuli evoked EPSCs with different interval. n = 9 neurons per group. The data were analyzed in separate *t*-tests at each stimulus intensity; n.s., p > 0.05. (**C**) AMPAR and NMDAR EPSCs in CA1 pyramidal neurons. Sample sweeps illustrating NMDAR-EPSCs recorded at +40 mV and AMPAR-EPSCs recorded at –70 mV (left). Histograms show ratio of NMDAR-EPSCs amplitude / AMPAR-EPSCs amplitude (right). n = 22 neurons per group; *p < 0.05, Student's *t*-test. (**D**) Sample traces and input-output curve of NMDA receptor-mediated EPSCs of control and sh*kif2c* neurons (n = 6, p > 0.05). The data were analyzed in separate t-tests at each stimulus intensity. (**E**) Sample traces and input-output curve of AMPA receptor-mediated EPSCs between control and sh*kif2c* neurons (n = 6, *p < 0.05). The data were analyzed in separate *t*-tests at each stimulus intensity. (**F**) Example EPSCs before (baseline) and after (t = 40 min) 2× HFS stimulation in control and sh*kif2c* CA1 pyramidal neurons. Time course of percentage changes of EPSCs amplitudes in control and sh*kif2c* CA1 pyramidal neurons(left). Histogram shows EPSC amplitude of control and sh*kif2c* neurons (35–40 min) (right). ***p < 0.001; Student's *t*-test.

The online version of this article includes the following source data for figure 3:

**Source data 1.** Values for mEPSCs of control and sh*kif2c* neurons.

various stimulation, while AMPAR-mediated EPSCs was increased (*Figure 5D–E*), indicating a specific defect in AMPAR-mediated synaptic currents. In addition, we performed cell-surface biotinylation experiments to examine the surface expression of these two receptors. The level of surface AMPA receptors was significantly increased in the cKO hippocampus (*Figure 5F*), which coincided with the electrophysiological abnormalities in AMPAR-mediated synaptic transmission in cKO mice. We also measured synaptic plasticity in the hippocampal CA1 region in WT and cKO mice. Result showed that the LTP magnitude was significantly attenuated in cKO hippocampal by two different induction protocols: single high-frequency stimulation (1× HFS) and 4-train LTP (4× HFS) (*Figure 5G and H*). Long-term depression (LTD) was not altered (*Figure 5I*). Taken together, these results suggest that KIF2C plays a critical role in LTP induction and maintenance.

## KIF2C regulates synaptic transmission and plasticity by mediating dynamic microtubule depolymerization in dendrites and spines

Extensive studies on KIF2C in dividing cells have shown that KIF2C regulates MT dynamics by mediating MT depolymerization (*Gorbsky, 2004*; *Helenius et al., 2006*; *Knowlton et al., 2006*). Moreover, dynamic MTs are reported to be involved in regulating dendritic spine morphology and synaptic

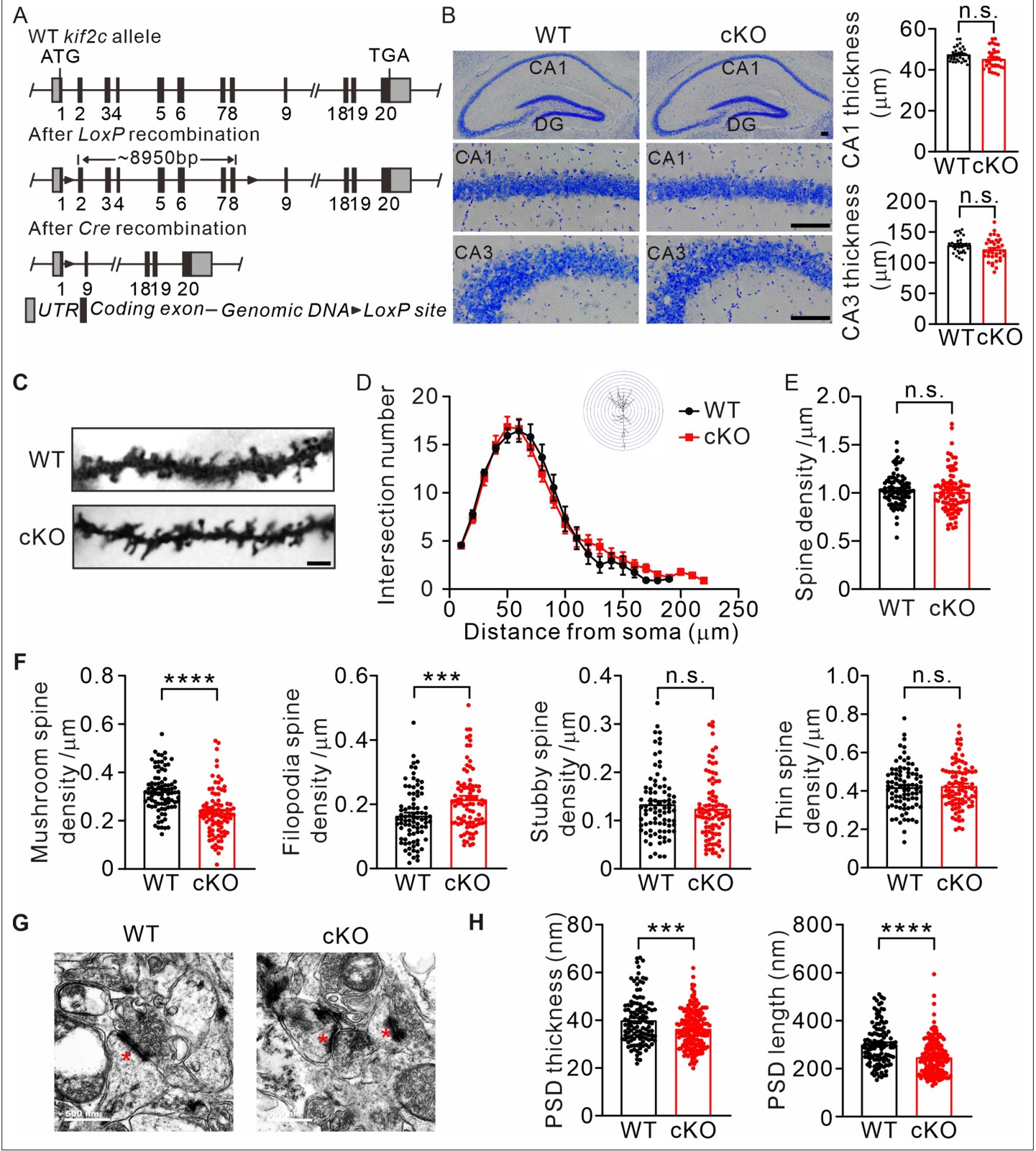

**Figure 4.** Abnormal spine formation in KIF2C conditional knockout mice. (**A**) Targeting strategy to generate *kif2c*<sup>flox/flox</sup> mice. Two LoxP sites were inserted into intron 1–2 and 8–9. (**B**) Nissl staining of adult mice brain coronal sections and magnified images of hippocampal CA1 and CA3 regions. Scale bar, 50 μm. n = 4 mice per genotypes; n.s., p > 0.05, Student's *t*-test. (**C–F**) Golgi staining of WT and cKO hippocampus CA1 pyramidal neuron. n = 4 mice per genotypes; Scale bar, 2 μm. (**D**) Sholl analysis of dendritic arborization. The number of intersections related to distance from soma was

*Figure 4 continued on next page*

*Figure 4 continued*

quantified. p > 0.05, Two-way ANOVA RM and *post hoc* comparisons. (**E**) Histograms show spine number per 1 µm of WT and cKO CA1 pyramidal neuron. p = 0.3809. Student's t-test. (**F**) The density of four types of spines in WT and cKO CA1 pyramidal neuron. Mushroom spine density was 0.32 ± 0.009 µm$^{-1}$ (WT) and 0.23 ± 0.01 µm$^{-1}$ (cKO) (****p < 0.0001). Filopodia spine density was 0.16 ± 0.009 µm$^{-1}$ (WT) and 0.22 ± 0.01 µm$^{-1}$ (cKO) (***p < 0.001). Stubby spine density was 0.12 ± 0.009 µm$^{-1}$ (WT) and 0.13 ± 0.01 µm$^{-1}$ (cKO) (p = 0.3104). Thin spine density was 0.41 ± 0.009 µm$^{-1}$ (WT) and 0.43 ± 0.01 µm$^{-1}$ (cKO) (p = 0.5511). Student's *t*-test. (**G**) Transmission EM of synapse in hippocampal CA1 region from WT and KIF2C cKO mice (n = 3 mice per genotype) at 6–8 weeks. Scale bar, 500 nm. (**H**) Quantification of PSD thickness and length. PSD thickness: 40.02 ± 0.93 nm (WT) and 36.37 ± 0.58 nm (cKO). PSD length: 301.4 ± 6.8 nm (WT) and 247.7 ± 5.4 nm (cKO). ***p < 0.001; ****p < 0.0001; Student's *t*-test.

The online version of this article includes the following source data and figure supplement(s) for figure 4:

**Source data 1.** Values for CA1 and CA3 thickness in WT and cKO mice; Values for Sholl analysis and spine density; Values for PSD thickness and length.

**Figure supplement 1.** *kif2c*$^{flox/flox}$;*Nestin*$^{Cre}$ mice.

**Figure supplement 1—source data 1.** Values for *kif2c* mRNA levels; Values for brain weight and body weight.

**Figure supplement 1—source data 2.** Original files of gels with the relevant bands.

plasticity (*Gu and Zheng, 2009*). Therefore, we speculated that KIF2C might regulate synaptic plasticity through adjusting dynamic MTs. To further explore this, we transfected EB3-tdtomato into WT or cKO hippocampal neurons to mark the MT plus end and observed the MT dynamics in mature neurons by live-cell imaging. Quantitative results showed that the speed of EB3 was increased in cKO dendrites (WT: 8.9 ± 0.8 µm / min, cKO: 12.4 ± 0.5 µm / min) (*Figure 6A and B*), indicating an increased rate of MT polymerization in cKO mice. Using the plusTipTracker software, which is designed to track MT growth parameters (*Myers et al., 2011*), we analyzed the polymerization properties of the EB3-labeled MT plus ends in dendrites. Depletion of KIF2C significantly increased MT plus-end growth speed (WT: 24.0 ± 1.1 µm / min, cKO: 31.0 ± 1.0 µm / min), and decreased its growth lifetime (WT: 9.4 ± 0.4 s, cKO: 6.7 ± 0.1 s) (*Figure 6B*). MT plus end growth length was not significantly altered (WT: 3.1 ± 0.1 µm, cKO: 3.0 ± 0.1 µm) (*Figure 6B*). These results suggest that KIF2C regulates MT depolymerization in hippocampal neurons and that MT dynamics become disordered with KIF2C deficiency.

MTs remain dynamic in mature neurons and are capable of invading dendritic protrusions, which depends on neuronal activity (*Hu et al., 2011*). To observe MT invasion, cultured hippocampus neurons were co-transfected with EB3-tdtomato (to label MT plus end) and GFP (to outline neuron volume and mark the spines). Quantitative results showed that MT spine invasion frequency was increased significantly (*Figure 6C*) in cKO neurons. We further treated neurons with 50 ng/mL of brain-derived neurotrophic factor (BDNF) to induce spine plasticity, and labeled newly polymerized MTs with an antibody against tyrosinated-α-tubulin. We found that without BDNF treatment, the population of spines containing tyrosinated-α-tubulin was larger in cKO neurons (*Figure 6D*), suggesting that KIF2C deficiency led to an increased MT spine invasion events under basal state. After BDNF treatment, MT spine invasion was robustly increased in WT neurons, while making no change in cKO neurons (*Figure 6D*), and such abolished augment could be rescued by re-expression GFP-KIF2C (*Figure 6E*), indicating that KIF2C is required for the activity-dependent MT invasion in spines.

In mitotic cells, KIF2C binds to the end of MTs and couples ATP hydrolysis to initiate MT depolymerization (*Sanhaji et al., 2011*). To further confirm whether the MT depolymerization property of KIF2C is critical for regulating synaptic transmission and plasticity, we constructed an MT depolymerization loss-of-function KIF2C mutant (KIF2C G491A). Its human homologous KIF2C G495A mutant has been previously reported to lose ATP hydrolyzing and hence MT depolymerization abilities (*Wang et al., 2012*; *Wang et al., 2015*). The KIF2C(G491A) mutant was first tested in HEK 293T cells for its MT depolymerization function. Immunocytochemical analysis showed that KIF2C(G491A) transfected cells had intact cell morphology, while the KIF2C(WT) successfully depolymerized MTs and maintained cell rounding (*Figure 6—figure supplement 1A*). Furthermore, we expressed KIF2C(WT) or mutant in the hippocampal CA1 region of cKO mice by AAV-mediated gene delivery (*Figure 6—figure supplement 1B*, C) and examined synaptic transmission and plasticity using an electrophysiological approach. The results showed that re-expression of KIF2C(WT) significantly reduced the elevated amplitude of mEPSCs (*Figure 6F*) and restored LTP in cKO mice (*Figure 6G*). However, without MT depolymerization activity, the KIF2C (G491A) mutant failed to show significant differences in mEPSC properties (*Figure 6F*) or LTP induction in the cKO CA1 neurons (*Figure 6G*). These findings indicate

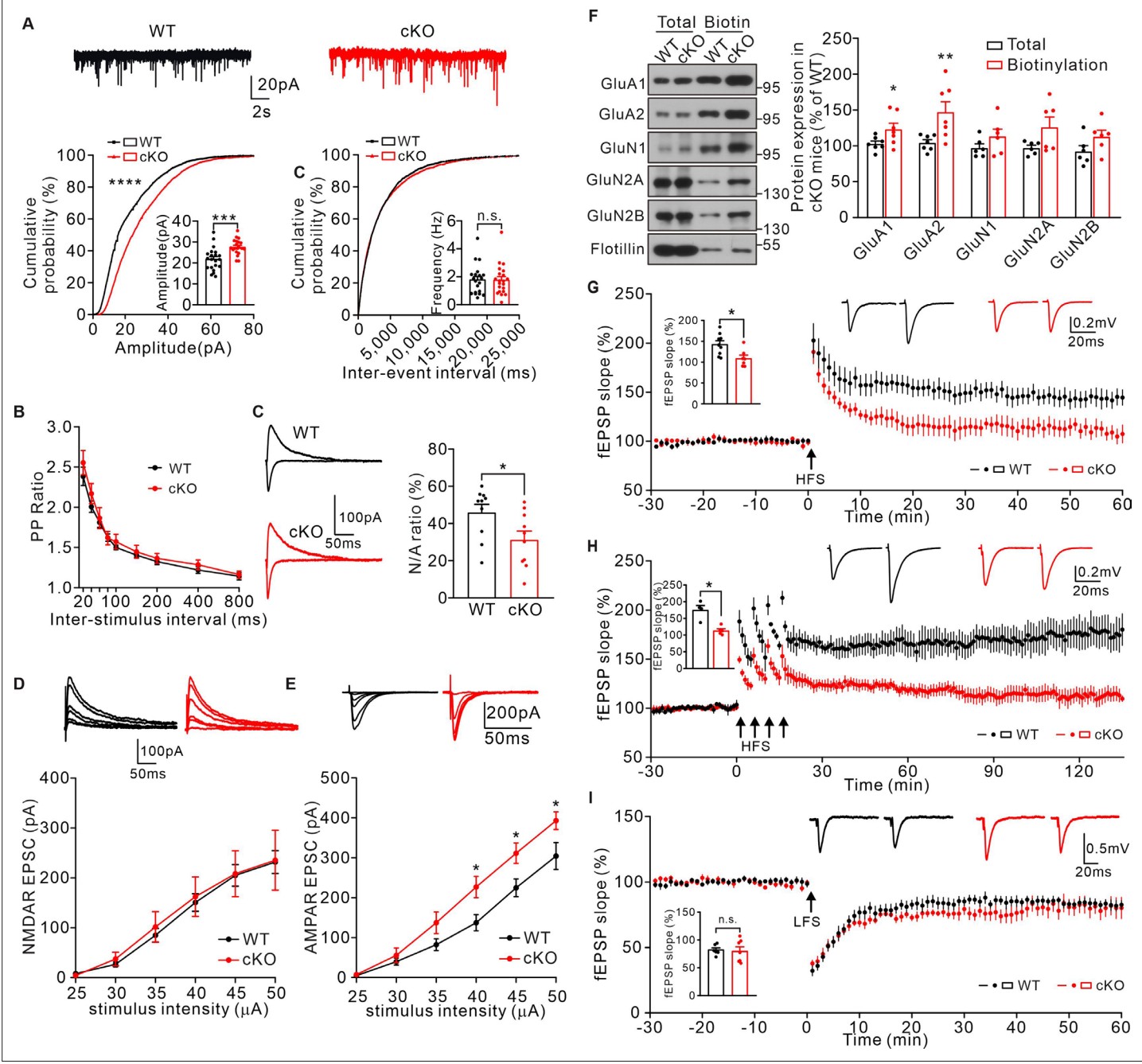

**Figure 5.** KIF2C deficiency impairs synaptic transmission and plasticity. (**A**) mEPSCs recorded from WT and cKO CA1 pyramidal neurons. n = 21 neurons per group; ***p < 0.001, Student's *t*-test; For the cumulative probability curves, n.s. p > 0.05, ****p < 0.0001, Kolmogorov-Smirnov test. (**B**) Paired-pulse stimuli evoked EPSPs with different interval. n = 9 neurons per group, p > 0.05. The data were analyzed in separate *t*-tests at each stimulus intensity. (**C**) AMPAR and NMDAR EPSCs in CA1 pyramidal neurons of WT and cKO mice. Sample sweeps illustrating NMDAR-EPSCs recorded at +40 mV and AMPAR-EPSCs recorded at −70 mV. Histograms show ratio of NMDAR-EPSCs amplitude / AMPAR-EPSCs amplitude (right). n = 10 neurons, *p < 0.05. Student's *t*-test. (**D**) Sample traces and input-output curve of NMDA receptor-mediated EPSCs between WT and cKO mice (n = 6 neurons, p > 0.05). The data were analyzed in separate *t*-tests at each stimulus intensity. (**E**) Sample traces and input-output curve of AMPA receptor-mediated EPSCs between WT and cKO mice (n = 6 neurons, *p < 0.05). The data were analyzed in separate *t*-tests at each stimulus intensity. (**F**) Cell-surface biotinylation shows a significant increase in GluA1 and GluA2 subunit surface expression in cKO hippocampus compared with WT. Histograms show percent change of GluA1, GluA2, GluN1, GluN2A, GluN2B subunits in total and biotinylation fraction. Flotillin was the internal control. n = 7 experiment repeats, *p < 0.05, **p < 0.01, Student's *t*-test. (**G**) Example fEPSP before (baseline) and after (t = 60 min) 1× HFS stimulation in WT and cKO CA1 slices. Time course of percentage changes of fEPSP slope in WT and cKO CA1 slices. WT, n = 9 slices from 7 mice; cKO, n = 7 slices from 6 mice. Histogram shows fEPSP slope of WT and cKO neurons (55–60 min). *p < 0.05, Student's *t*-test. (**H**) Example fEPSP before (baseline) and after (t = 135 min) 4× HFS stimulation in

*Figure 5 continued on next page*

*Figure 5 continued*

WT and cKO CA1 slices. Time course of percentage changes of fEPSP slope in WT and cKO CA1 slices. WT, n = 4 slices from 4 mice; cKO, n = 5 slices from 4 mice. Histogram shows fEPSP slope of WT and cKO neurons (130–135 min). *p < 0.05, Student's *t*-test. (I) Time course of percentage changes of fEPSP slope before (baseline) and after low frequency stimulus (LFS) in control and cKO CA1 slices. WT, n = 7 slices from 5 mice; cKO, n = 7 slices from 6 mice. Histogram shows fEPSP slope of WT and cKO neurons (55–60 min). p = 0.7642, Student's *t*-test.

The online version of this article includes the following source data for figure 5:

**Source data 1.** Values for mEPSCs of WT and cKO neurons.

**Source data 2.** Original files of blots with the relevant bands.

that KIF2C regulates synaptic transmission and plasticity by mediating dynamic MT depolymerization in dendrites and spines.

## KIF2C plays an important role in CA1-dependent cognitive behaviors

KIF2C deficiency leads to abnormalities in synaptic transmission and synaptic plasticity; we speculate whether KIF2C plays a role in higher order brain function. Therefore, we examined KIF2C cKO mice for several behavioral tests. The KIF2C cKO mice showed normal motion and anxiety levels in the open field test and elevated zero maze compared to those of the WT mice (*Figure 7—figure supplement 1A, B*), and had normal grooming frequencies in the commonly used repetitive behavioral test (*Figure 7—figure supplement 1C*). The Morris water maze task also elicited similar learning curves and probe performance in both KIF2C cKO and WT mice (*Figure 7—figure supplement 1D*), suggesting normal spatial learning and memory ability in mice with KIF2C deficiency. However, when we subjected the mice to the Y-maze test, KIF2C cKO mice showed significant impairment in spontaneous alternation behavior compared to that of the WT mice (*Figure 7A*), indicating that KIF2C cKO mice were deficient in short working memory, which could be due to the special role of CA1 in temporal processing (*Hoge and Kesner, 2007*; *Ji and Maren, 2008*; *Farovik et al., 2010*).

To investigate whether this learning deficit would translate to other cognitive disorders, we examined the mice for cued conditioned fear memory and social memory. The conditioned fear learning test showed that KIF2C cKO mice displayed a comparatively weaker freezing response to the context and tone conditioned stimulus (CS) during the acquisition stage (*Figure 7B*). We monitored the conditioned freezing response to the background context and CS until seven or 8 days after acquisition, respectively, and found that cKO mice still showed less freezing time compared to WT mice, particularly in context-dependent freezing (*Figure 7C*). These results suggest that fear memory in KIF2C cKO mice was impaired. The percentage change in freezing time between the KIF2C WT and cKO mice showed no significant difference, indicating an intact fear memory maintenance in cKO mice (*Figure 7—figure supplement 1E*). In a prepulse inhibition test, WT and cKO mice showed the same extent of inhibition as the prepulse (*Figure 7—figure supplement 1F*), showing that the cKO mice had normal sensorimotor gating. In the three-chambered social test, both WT and KIF2C cKO mice exhibited no position preference in habituation stage (*Figure 7—figure supplement 1G*). KIF2C cKO mice exhibited a normal preference for the chamber in which a stranger mouse (S1) was present, compared with the empty chamber (*Figure 7D*). However, with the introduction of a second stranger (S2), KIF2C cKO mice exhibited no preference for the novel stranger (S2), compared with the familiar mouse (S1) (*Figure 7E*). These results showed that KIF2C cKO mice showed normal sociability but significantly reduced social novelty, indicating that KIF2C is essential for social memory maintenance.

Based on the behavioral tests, we found that KIF2C plays an important role in CA1-mediated cognitive and memory functions. The deficiencies observed in the Y-maze test, cued conditioned fear memory, and three-chambered social test of KIF2C cKO mice were highly likely due to the lack of LTP in the CA1 region (*Figure 5G and H*). As re-expression of KIF2C(WT) in the CA1 region restored LTP (*Figure 6G*), we speculated whether such CA1-specific rescue could also ameliorate the behavioral abnormalities in KIF2C cKO mice. In our pretests, over-expression of KIF2C in mice for a long time would lead to neuronal death (data not shown). Therefore, we re-expressed KIF2C in the cKO mice CA1 region using a doxycycline-induced viral expression system (*Figure 6—figure supplement 1C*, *Figure 7—figure supplement 1H*) and examined the rescuing effects in the Y-maze test and three-chambered social test. The results showed that cKO mice with KIF2C(WT) re-expression displayed a normal alternation in the Y-maze test compared to cKO mice with control virus (*Figure 7F*,

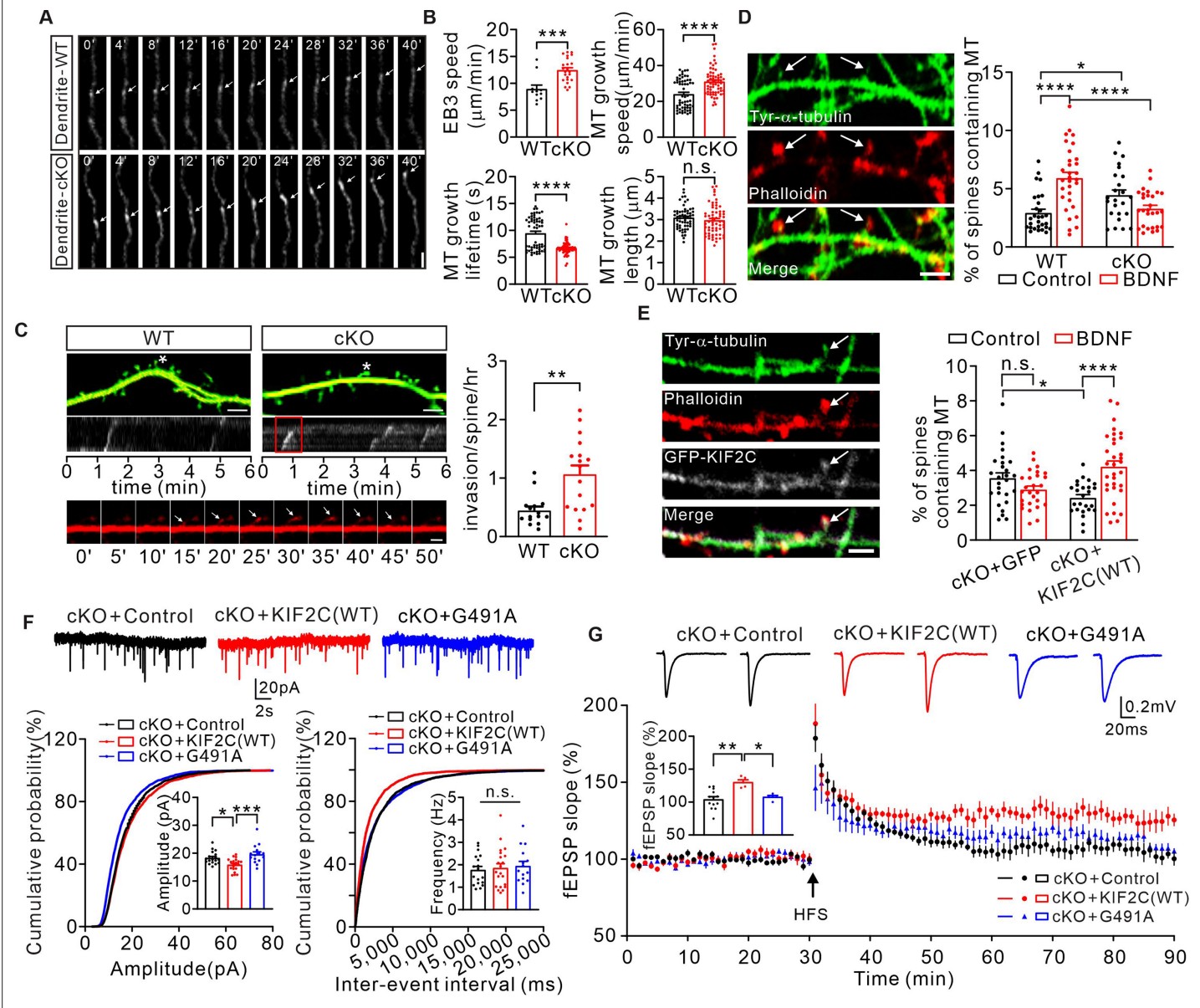

**Figure 6.** Abnormal MT dynamics causes deficit in synaptic transmission and plasticity. (**A**) Time-lapse recordings of EB3-tdtomato-infected neurons showing EB3-tdtomato comets in dendrite of WT and cKO hippocampus neuron. Subsequent images show low-pass-filtered time series in row. Scale bar is 1 μm, time in seconds. (**B**) Analysis of the EB3 comet dynamics in dendrites of WT and cKO hippocampus neuron. n = 58 per group from seven independent experiments. Histograms show EB3 speed velocity (***p < 0.001), MT growth speed (****p < 0.0001), MT growth lifetime (****p < 0.0001) and MT growth length (p = 0.2850). Student's t-test. (**C**) Images of dendrites from WT and cKO cultured hippocampal neurons transfected with EB3-tdtomato (red) and GFP (green). Scale bar, 5 μm. Spines labeled with "*" were depicted in the kymograph. The invasion shown in the red boxed region of kymograph is depicted below. Sequential frames showed an MT entering the labeled spine of cKO neuron. Scale bar,1 μm. Histograms show MT-spine invasion frequency (invasions/spine/hour). (WT: 0.45 ± 0.07, n = 15; cKO: 1.07 ± 0.15, n = 17. **p < 0.01). Student's t-test. (**D**) Tyrosinated α-tubulin antibody-labeled MTs (white arrows) were detected in a small percentage of phalloidin (F-actin) -labeled spines under basal conditions (DIV 18–20). Quantification of the number of spines containing MTs before and after BDNF treatment. Scale bar, 5 μm. 2.9% ± 0.32% (WT, n = 29), 5.9% ± 0.5% (WT+ BDNF, n = 31), 4.5% ± 0.46% (cKO, n = 24), 3.3% ± 0.29% (cKO+ BDNF, n = 30). *p < 0.05, ****p < 0.0001, One-way ANOVA (post hoc comparison). (**E**) DIV 18–20 cKO hippocampus neurons transfected with GFP vector or GFP-KIF2C labeled with GFP (white), tyrosinated α-tubulin (green) and phalloidin (red). Quantification of the number of spines containing MTs before and after BDNF treatment. Scale bar, 2 μm. 3.6% ± 0.31% (cKO+ GFP + control, n = 27), 2.9% ± 0.20% (cKO+ GFP + BDNF, n = 27), 2.4% ± 0.19% (cKO+ KIF2 C(WT)+ control, n = 25), 4.2% ± 0.34% (cKO+ KIF2 C(WT)+ BDNF, n = 33). *p < 0.05, ****p < 0.0001, One-way ANOVA (post hoc comparison). (**F**) mEPSCs recorded from control (n = 18 neurons), KIF2C(WT) (n = 21 neurons) and KIF2C(G491A) (n = 15 neurons) expressed in cKO CA1 pyramidal neurons. n.s. p > 0.05, *p < 0.05, ***p < 0.001, One-way ANOVA (post hoc comparison). (**G**) Example fEPSP before (baseline) and after (t = 60 min) 1× HFS stimulation in cKO+ control, cKO+ KIF2 C(WT) and cKO+

*Figure 6 continued on next page*

Figure 6 continued

KIF2 C(G491A) CA1 slices. Time course of percentage changes of fEPSP slope in cKO+ control, cKO+ KIF2 C(WT) and cKO+ KIF2 C(G491A) CA1 pyramidal neurons. cKO+ control, n = 13 slices from 10 mice; cKO+ KIF2 C(WT) and cKO+ KIF2 C(G491A), n = 5 slices from 4 mice. Histogram shows fEPSP slope of cKO+ control, cKO+ KIF2 C(WT) neurons and cKO+ KIF2 C(G491A) (55–60 min). *p < 0.05, **p < 0.01; One-way ANOVA (*post hoc* comparison).

The online version of this article includes the following source data and figure supplement(s) for figure 6:

**Source data 1.** Values for EB3 speed velocity, MT growth speed, MT growth lifetime, and MT growth length.

**Figure supplement 1.** KIF2C(WT) and KIF2C(G491A) expression.

**Figure supplement 1—source data 1.** Original files of blots with the relevant bands.

*Figure 7—figure supplement 1I*). Furthermore, cKO mice with KIF2C(WT) re-expression also showed normal interaction with the stranger mouse in the social novelty test (*Figure 7G–H*). In addition, we tested whether the expression of the functional-null mutant KIF2C(G491A) in the CA1 region of cKO mice would have any rescue effect. The results showed that cKO mice injected with KIF2C(G491A) virus still showed impaired working and social memory (*Figure 7F–H*, *Figure 7—figure supplement 1I*). Therefore, these results suggest that KIF2C is critical for cognitive function, and this functional role is mediated by its MT depolymerization ability in synapses during LTP.

## Discussion

KIF2C is an ATP-dependent MT depolymerization factor that functions in several mitotic processes, such as spindle assembly, MT dynamics, correct kinetochore-MT attachment, and chromosome positioning and segregation (*Sanhaji et al., 2011*). However, its function in the CNS was largely unknown. Our study elucidated the functions of KIF2C in regulating spine morphology, synaptic transmission, and plasticity in hippocampal neurons and its critical role in CA1-mediated cognitive behaviors. Our study unveils the critical role of KIF2C in the CNS, showing that its known function in regulating dynamic MT stability also takes place in neurons.

The stabilization of dendritic spines is an important structural basis for synaptic plasticity. According to our results, KIF2C deficiency seems to have higher impact on synaptic structure plasticity than that on spine formation, which provides a link between dynamic MT stability and synaptic plasticity. We have also found that both acute knockdown or genetic deficiency of KIF2C eliminated LTP but not LTD expression. On the molecular basis, KIF2C knockdown or cKO hippocampal neurons exhibited a higher AMPAR-mediated EPSC amplitude and an increase in surface AMPARs expression. It is possible that increased expression of ionic glutamate receptors, particularly AMPARs, on synapses leads to LTP occlusion. Moreover, 4-train LTP impairment in cKO mice, which requires further gene transcription and protein synthesis in the postsynaptic neuron, suggests that KIF2C might also be involved in the expression and/or transportation of synaptic substances for LTP long-lasting maintenance. It would be interesting to further explore the mechanism of KIF2C-involved receptor trafficking or synaptic expression.

As important architectural elements in neurons, MTs are also responsible for intracellular transport. It has been reported that NMDA receptors can be transported by KIF17 (*Setou et al., 2000*) and AMPA receptors can be transported by KIF5 along MTs in dendrites (*Setou et al., 2002*). It is possible that KIF2C regulates membrane receptor trafficking by mediating MT dynamics through the regulation of MT stability. We observed MT dynamics using live cell imaging of KIF2C cKO hippocampal neurons. We observed that depletion of KIF2C induced faster, shorter lived MT growth in dendrites. KIF2C deficiency also increased the frequency of MT invasion under basal conditions, which was probably due to the loss of MT polymerization suppression. These results suggest that KIF2C regulates MT depolymerization and stability in hippocampal neurons. Furthermore, MT dynamic stability regulates AMPARs postsynaptic localization and synaptic plasticity (*McVicker et al., 2016*; *Uchida et al., 2014*). In the case of AMPARs trafficking, MT-dependent trafficking has been typically associated with long-range transport along the dendrites, supporting the continuous supply and redistribution of AMPARs among synapses (*Hoerndli et al., 2013*; *Heisler et al., 2014*), which can also be regulated by neuronal activity (*Hoerndli et al., 2013*). We speculate that KIF2C may also confine the AMPARs' trafficking to the synaptic site through MT depolymerization and stability.

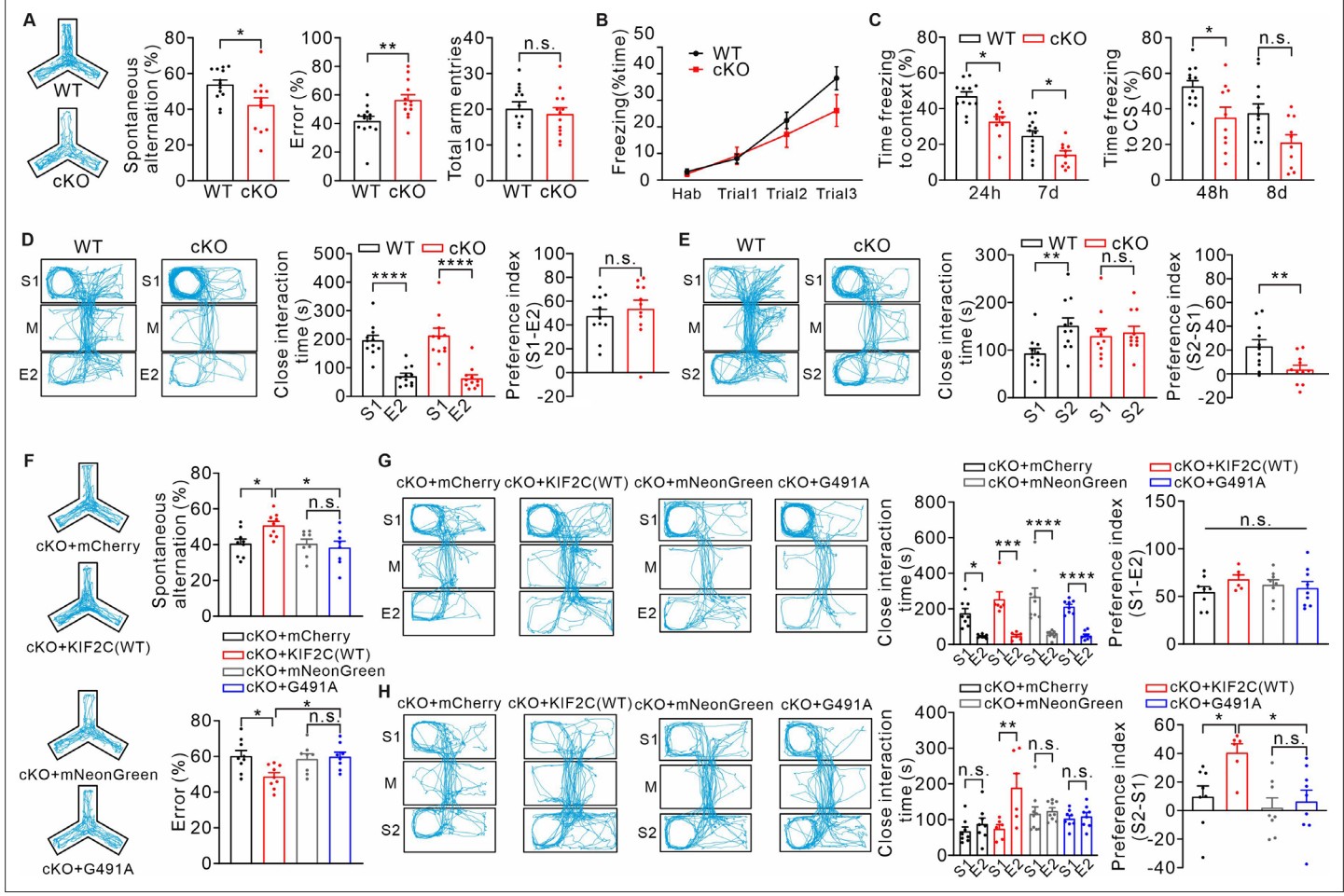

**Figure 7.** KIF2C cKO mice exhibit abnormal cognitive behaviors. (**A**) Y maze test from WT and cKO. n = 13 per genotype. n.s., p > 0.05, *p < 0.05, **p < 0.01, Student's *t*-test. (**B–C**) Cued fear conditioning test. (**B**), WT and cKO mice showed no significant differences of freezing response to the tone across three conditional stimulus (CS)-unconditional stimulus (US) pairings. p > 0.05; Two-way ANOVA. (**C**), Expression of conditioned freezing response to the background context or CS at indicated time. WT: n = 12; cKO: n = 10. *p < 0.05, Two-way ANOVA RM comparisons. (**D–E**) Three-chamber test. (**D**), WT and cKO mice spent more time with stranger one in sociability test. n.s., p > 0.05, ****p < 0.0001. Student's *t*-test. (**E**), cKO mice did not display a preference for stranger two in social novelty test. n = 11 per genotype. n.s., p > 0.05, **p < 0.01, Student's *t*-test. (**F**) cKO mice with KIF2C(WT) re-expression displayed a normal alternation in the Y-maze test. cKO mice with KIF2C(G491A) re-expression displayed similar deficiency in the Y-maze test. cKO+ mCherry, n = 9; cKO+ KIF2 C(WT), n = 9; cKO+ mNeonGreen, n = 8; cKO+ KIF2 C(G491A), n = 8. n.s., p > 0.05, *p < 0.05, One-way ANOVA (*post hoc* comparison). (**G–H**) KIF2C(WT) re-expression in CA1 can restore social novelty in cKO mice. cKO mice with KIF2C(G491A) expression displayed similar deficiency in social novelty test. cKO+ mCherry: n = 8; cKO+ KIF2 C(WT): n = 6; cKO+ mNeonGreen: n = 8; cKO+ KIF2 C(G491A): n = 9. n.s., p > 0.05, *p < 0.05, ***p < 0.001. One-way ANOVA (*post hoc* comparison).

The online version of this article includes the following source data and figure supplement(s) for figure 7:

**Source data 1.** Values for Y-maze test of WT and cKO mice.

**Figure supplement 1.** WT and KIF2C cKO mice behavioral tests.

**Figure supplement 1—source data 1.** Values for moving distance, velocity and center zone exploration time in the open-field test.

Activity-dependent MT invasion of dendritic spines becomes disabled after BDNF stimulation in cKO hippocampal neurons. Our findings reveal that during cLTP induction, KIF2C was removed from the synapses (*Figure 2F–I*). It is possible that translocation of KIF2C out from synapse or depletion of synaptic KIF2C during synaptic activation is required to ensure robust MT invasion into spines, where and MT depolymerization should be inhibited. However, we still need direct evidence to prove this hypothesis. Even though the genetic depletion of KIF2C increases the basal MT invasion into spines, and largely prevents further increase in MT invasion after BDNF stimulation, we are not clear whether this is the result of depletion of KIF2C's function in spines or a followup consequence after dendritic failure of MT depolymerization, that is, after depletion of KIF2C's function in dendrites.

KIF2C's translocation after cLTP induction would be a touch to its pure role in spines, and it would be interesting to further explore the underlying mechanism.

Abnormal dynamic MT stability would further lead to synaptic plasticity failure and, finally, cognition defects. This hypothesis was further supported by our rescue experiments using the KIF2C(G491A) mutant, in which ATP hydrolyzing and MT depolymerization capabilities are abolished (*Wang et al., 2015*). Re-expression of KIF2C(WT), but not KIF2C(G491A) mutant, rescued the mEPSC hyper-amplitude in KIF2C cKO mice. KIF2C(G491A) also failed to restore the impaired LTP or cognitive function in KIF2C cKO mice during working memory or social memory tests. Therefore, the MT depolymerization function of KIF2C plays a critical role in regulating synaptic plasticity and cognitive behaviors.

Interestingly, Oligomeric amyloid-β directly inhibits the MT-dependent ATPase activity of KIF2C in vitro, and thereby leads to the generation of defective mitotic structures which may contributes to the development of Alzheimer's disease (*Borysov et al., 2011*). Aβ stabilizes MT by reducing catastrophe-frequency (*Qu et al., 2017*), which is similar with KIF2C conditional knockout phenotype. Furthermore, KIF2C was recently discovered as a blood biomarker for suicidal ideation in psychiatric patients, such as patients with schizophrenia (*Niculescu et al., 2015*).Therefore, KIF2C could be a therapeutic target for Alzheimer's disease or suicidality psychiatric disorders. Nevertheless, whether KIF2C plays other important roles in higher-order brain functions remain to be explored.

## Materials and methods

Generation of *kif2c* *flox/flox* mice and animal maintenance *kif2c*$^{flox/flox}$ mouse was generated via CRISPR/Cas9 system in the Model Animal Research Center of Nanjing University (Nanjing, China). In brief, Cas9 mRNA, sgRNA and donor were co-injected into zygotes. sgRNA-5′ end: 5′- TCC ACT GTG GAA TGG TCT AA –3′; sgRNA-3′ end: 5′- CTG GTT GTG AGA CAC CTG TA-3′. Mouse *kif2c* gene is composed of 20 exons. The sgRNA directs Cas9 endonuclease cleavage in intron 1–2 and intron 8–9 and create a DBS (double-strand break). DBS breaks were repaired by homologous recombination, and resulted in LoxP sites inserted in intron 1–2 and intron 8–9 respectively. The mouse pups were genotyped by PCR and Southern blot followed by sequence analysis. Conditional knockout mice (*kif2c*$^{flox/flox}$;*Nestin*$^{Cre}$) were obtained by crossing *kif2c*$^{flox/flox}$ mice with *Nestin*$^{Cre}$ mice (*Wang et al., 2019*). The resulting offspring were genotyped by PCR of genomic DNA. The primers information: *kif2c* floxP fragment, F: 5′-GGT CCA GCT CTT TAC TGA TGT GTT C –3′; R: 5′-ACA AAG CAA GTC CAG GTC CAA G-3′; *Nestin-Cre*, F: 5′-TGC AAC GAG TGA TGA GGT TC-3′; R: 5′- GCT TGC ATG ATC TCC GGT AT-3′. Mice were kept under temperature-controlled conditions on a 12:12 hr light/dark cycle with sufficient food and water ad libitum. Experiments were done in male littermates at the age of 1~3 month. All animal experiments were performed in accordance with the ethical guidelines of the Zhejiang University Animal Advisory Committee (2020–19500#), and were in complete compliance with the National Institutes of Health Guide for the Care and Use of Laboratory Animals. All surgeries were performed under sodium pentobarbital anesthesia, and every effort was made to minimize suffering.

### Antibodies and reagents

Antibody to KIF2C (12139–1-AP, RRID:AB_2877829) was purchased from Proteintech (Rosemont, IL). The anti-PSD-95 (ab2723, RRID:AB_303248), anti-MAP2 (ab5392, RRID:AB_2138153), anti-α-tubulin antibody (ab52866, RRID:AB_869989), anti-mCherry (ab167453, RRID:AB_2571870), anti-GFP (ab6556, RRID:AB_305564) and anti-EB3 (ab157217, RRID: AB_2890656) were purchased from Abcam (Cambridge, UK). The anti-α-tubulin, tyrosinated (MAB1864-I, RRID:AB_2890657), anti-synaptophysin (MAB368, RRID:AB_94947), anti-GAPDH (MAB374, RRID:B_2107445), anti-GluA1(04–855, RRID:AB_1977216), and anti-GluA2 (MAB397, RRID:AB_2113875) were purchased from Millipore (Billerica, MA). The anti-GluN2A (4205, RRID:AB_2112295), anti-GluN2B (4207, RRID:AB_1264223), anti-GluN1 (5704, RRID:AB_1904067) were from Cell Signaling Technology (Danvers, MA). The anti-flotillin was from BD (610820, RRID:AB_398139). The anti-β-tubulin (sc-5274, RRID:AB_2288090) was from Santa Cruz (Dallas, TX). The anti-bassoon (ADI-VAM-PS003-F, RRID:AB_11181058) was from Enzo Life Sciences (Farmingdale, NY). Anti-N-cadherin Antibody (BS2224, RRID:AB_1664028) was purchased from Bioworld Technology (Bloomington, MN, USA). The Phalloidin was from Yeasen (40,734ES75, Shanghai, China). Horseradish peroxidase-conjugated secondary antibodies for immunoblotting were

from Jackson and for immunostaining were from Invitrogen (Carlsbad, CA). Dulbecco's modified Eagle's medium (DMEM), Neurobasal medium, B27 were from Gibco. GlutMax, 4',6-diamidino-2-phenylindole (DAPI), and Alexa Fluor-conjugated secondary antibodies were from Invitrogen (Carlsbad, CA). Nissl was from Beyotime (Shanghai, China). Protease inhibitor cocktail was from Merck Chemicals. Other chemicals were from Sigma (St. Louis, MO) unless stated otherwise.

## qRT-PCR

For single-cell analysis, the tip of a conventional patch-clamp pipette was placed tightly on the soma of a selected individual CA1 neuron. Gentle suction was applied to the pipette. After complete incorporation of the soma, the negative pressure was released and the pipette was quickly removed from the bath. The harvested contents were subjected to qRT-PCR using OneStep Kit (Qiagen, Hilden, Germany).

For tissue analysis, the hippocampus tissue RNA was extract by using RNeasy Mini Kit (Qiagen, Hilden, Germany). In brief, hippocampus tissue was homogenized in buffer RLT. The lysate was centrifuged at 12,000× g (4 °C for 3 min). The supernatant was transferred and 1 volume of 70% ethanol was added. Then RNA was absorbed on RNeasy Mini spin column and purified by washing with buffer RW1 and RPE. The harvested contents were subjected to Reverse Transcription Kit (Vazyme). The qPCR reaction was performed on HiScript II 1st Strand cDNA Synthesis Kit (R212-02) (Vazyme, Nanjing, China) using ChamQ Universal SYBR qPCR Master Mix (Vazyme, Q711). Primer information: *gapdh*: F 5'- AGG TCG GTG TGA ACG GAT TTG-3'; R 5'- TGT AGA CCA TGT AGT TGA GGT CA-3'. *kif2c*: F 5'-TGC CGT TGT TGA TGG TCA GTG-3'; R 5'-GGA GAC ACT TGC TGG GAA CAG-3'. sh*kif2c*: F 5'-TGG ATC GAA GGA GGT ACC AC-3'; R 5'- CAC TGA CCA TCA ACA ACG GCA-3'. The following parameters were used: 95 °C for 30 s, followed by 39 cycles of 95 °C for 10 s, 60 °C for 30 s in accordance with the manufacturer's protocol.

## Hippocampal neuron culture and transfection

Primary cultures of hippocampal neurons were prepared from embryonic day 18 mice. The isolated hippocampi were dissociated with 2.5% trypsin at 37 °C for 20 min. The neurons were quantified and centrifuged at 1800 rpm for 6 min and then plated on the glass coverslips pre-coated with poly-L-lysine (Sigma). The culture medium was Neurobasal (Gibco) supplemented with 2% B27 (Gibco), GlutaMax (Invitrogen) and 1% Penicillin-Streptomycin (Life Technologies) in a 5% $CO_2$ atmosphere at 37 °C. Neurons were transfected at DIV 7–10 using a modified calcium phosphate protocol (*Jiang and Chen, 2006*). More than 10 days after transfection, neurons were directly subjected to live imaging under a Nikon A1R confocal scanning microscope.

## HEK 293T culture and transfection

HEK 293T cell line was purchased from ATCC (CRL-3216) through Chinese Academy of Sciences Stem Cell Bank (Serial number: SCSP-502). A spin-off strain with high transfection efficiency produced after the 293 cell line is inserted into the temperature-sensitive gene of the SV40 T-antigen. Mycoplasma test result was negative. The results of STR identification are: D5S818: 8,9; D13S317: 12,14; D7S820: 11; D16S539: 9,13; vWA: 16,19; THO1: 7,9.3; Amelogenin: X; TPOX: 11; CSF1PO: 11,12. HEK 293T cells were cultured in MEM media (Gibco) containing 10% fetal bovine serum (Gibco) and Penicillin-Streptomycin (Life Technologies). HEK 293T cells were transfected using the lipofectamine 2000.

## DNA constructs

The shRNA targeting sequence is designed against position 820–838 of mouse *kif2c* (Gene ID: 73804) open reading frame (5'-CCGGATGATCAAAGAATTT-3'), using Bgl Ⅱ/Sal Ⅰ restriction enzymes. GFP-KIF2C(WT) used the Nhe Ⅰ/BamH Ⅰ restriction enzymes. GFP-KIF2C(G491A) point mutations were replaced a G to A at 491 position, using the Nhe Ⅰ/BamH Ⅰ restriction enzymes. Plasmid synthesis and virus construction were completed by the company (OBIO, Shanghai, China). KIF2C shRNA was sub-cloned into AAV backbone pAKD-CMV-bGlobin-mcherry-H1-shRNA. The KIF2C shRNA lentivirus used for infecting primary hippocampal neurons were prepared as described (*Xu et al., 2016*). KIF2C(WT) was sub-cloned into AAV backbone pAAV-CMV-MCS-3FLAG-P2A-mNeonGreen-CW3SL (OBIO, Shanghai, China) and rAAV-TRE3G-MCS-3*Flag-2A-mCherry-WPRE-pA (BrainVTA,

Wuhan, China). KIF2C(G491A) was sub-cloned into AAV backbone pAAV-CMV-MCS-3FLAG-P2A-mNeonGreen-CW3SL (OBIO, Shanghai, China).

## Live-cell imaging and microtubule dynamics analysis

The hippocampus neurons were cultured in the glass bottom dish (Cellvis, D35-20-1-N) pre-coated with poly-L-lysine, transfected by plasmids at DIV 7–10 and imaged at DIV 18–20. Live-cell imaging of EB3-tdtomato comets were captured with Nikon A1R confocal microscope with perfect focus system. Neurons were kept at 37 °C in a Nikon incubation chamber with 5% $CO_2$ concentration and humidity during time-lapse imaging. All images were collected at 1 s time intervals, for total 10 min durations. EB3-tdtomato comets polymerization speed, lifetime and length were measured by plusTipTracker software. The following tracking settings were used: camera bit depth, 16 bits; detection method, watershed-based method with $\sigma_1 = 1$ and $\sigma_2 = 4$ and $K = 5$. Parameters used for comet tracking were as follows: search radius range, 3–15 pixels; minimum sub-track length, 3 frames; maximum gap length, 3 frames; maximum shrinkage factor, 1.5; maximum forward angle, 30°; maximum backward angle, 10°; fluctuation radius, 1.25 pixels (*Lazarus et al., 2013*; *Sánchez-Huertas et al., 2020*). Kymographs were created from lines drawn along the length of invaded spines. Invasion frequency, defined as the number of MT invasions per spine per hour, was calculated for each neuron by counting the total number of MT-spine invasions during 10 min.

## Adeno-associated virus microinjection

For virus microinjection, WT or KIF2C cKO mice were randomly allocated to experimental groups and processed. A small craniotomy was performed after anesthetizing the mouse. The flow rate (40 nl/min) was controlled by a micro-injector (World Precision Instruments). Virus was microinjected in both left and right hippocampal CA1 region (from bregma, 3 weeks: ML: 1.45 mm; AP: 1.8 mm; DV: 1.5 mm; 2 months: ML: 1.65 mm; AP: 1.82 mm; DV: 1.62 mm). For electrophysiology, 4- to 6-week-old mice were subjected for experiments at least 3 days after injection. For KIF2C re-expression mice subjected for behavioral tests, mice were injected and bred until 2- to 3-month-old, then took water containing (40 µg/ml) doxycycline for 2.5–3 days to induce KIF2C expression before tests.

## Glycine-induced LTP

Hippocampal neurons were infected by shRNA virus at DIV 7–10, and subjected for chemical LTP induction at DIV 18–20. Briefly, neurons were firstly incubated in extracellular solution (ECS; in mM: 140 NaCl, 2 $CaCl_2$, 5 KCl, 5 HEPES, 20 glucose) supplemented with 500 nM TTX, 1 µM strychnine, 20 µM bicuculline (Buffer A; pH 7.4) at 37 °C for 5 min, and treated with 200 µM glycine in Buffer A for 10 min. Then neurons were incubated in Buffer A for another 5–10 min. The cells were then harvested for experiments (*Lu et al., 2001*; *Merriam et al., 2013*; *Zhang et al., 2015*).

## Western blots

After determining protein concentration with BCA protein assay (Thermo, 23225), equal quantities of proteins were loaded and fractionated on SDS-PAGE and transferred to NT nitrocellulose membrane (Pall Corporation), immunoblotted with antibodies, and visualized by enhanced chemiluminescence (Pierce Biotechnology). Primary antibody dilutions used were GluA1 (1:1000), GluA2 (1:2000), GluN1 (1:2000), GluN2A (1:2000), GluN2B (1:2000), flotillin (1:1000), KIF2C (1:500), EB3 (1:10,000), PSD-95 (1:1000), synaptophysin (1:2000), β-tubulin (1:2000), N-cadherin (1:1000), mCherry (1:2000), GAPDH (1:10,000), and secondary antibodies (1:10,000). Film signals were digitally scanned and quantitated using Image J software.

## Immunocytochemistry and immunofluorescence quantification

DIV 18–20 hippocampal neurons were fixed in 4% paraformaldehyde (PFA) containing 4% sucrose for 20 min, and then treated with 1% Triton X-100 in PBS for 10 min at room temperature (RT). After washing with 1× PBS, neurons were transferred into blocking solution (10% normal donkey serum, NDS in PBS) for 1 hr at RT. Neurons were then incubated with primary antibodies at 4 °C overnight and incubated with secondary antibodies for 3 hr at RT. Primary antibody dilutions used for immunocytochemistry were PSD-95 (1:1000), KIF2C (1:500), bassoon (1:1000), GFP (1:2000), tyrosinated α-tubulin (1:500), α-tubulin (1:1000) and MAP2 (1:10,000), and secondary antibodies (1:1000). All

antibodies were diluted in 1× PBS containing 3% NDS. All images were collected by confocal laser scanning microscope Nikon A1R. Measures of spine head width and spine density in cultured hippocampal neurons were performed using Imaris software and colocalization analysis was performed by Metamorph software. Generally, fluorescence intensity was measured using region measurements function in MetaMorph. Single dendrites were selected randomly. A threshold value was set manually and separately to optimize the representation of puncta in green (PSD-95) and red (KIF2C) channels and then applied to all images from the same experiment. For the colocalization quantification, the area percentage of KIF2C overlapping PSD-95 was performed using Measurement Colocalization function. Dendritic shaft areas were manual selected and the shaft KIF2C intensity were calculated. For the quantification of synaptic KIF2C density and intensity, all the puncta on the green (PSD-95) channel were selected and used as a mask for red channel (KIF2C) overlapping to filter out the PSD-95-colocalized KIF2C puncta automatically, using the Process function, and Integrated Morphometric Analysis function was used to identify the puncta and measure their properties.

## Immunohistochemistry

Mice were anesthetized and transcranially perfused with 4% PFA. Coronal slices with 30 μm thickness were prepared and placed in blocking solution for 1 hr at RT. After washing with 1× PBS, slices were incubated with primary antibodies overnight at 4 °C and incubated with secondary antibodies for 3 hr at RT. Images were acquired using an automated fluorescence microscope Olympus BX53.

## Nissl staining

Nissl staining was performed using Nissl staining Kit (Beyotime, C0117). Coronal slices (30 μM thickness) were immersed in Nissl staining solution for 10 min at 37 °C, rinsed with ddH$_2$O, dehydrated in ethanol, and cleared in xylene. Images of cortex and hippocampus were captured using an Olympus VS120 microscope. Hippocampus CA1 and CA3 thickness was calculated by Image J software.

## Golgi staining

Golgi-Cox staining of hippocampus CA1 tissue was performed using the FD Rapid GolgiStain Kit (FD NeuroTechnologies, PK401) following the manufacturer's protocol. In brief, mice were deeply anesthetized with intraperitoneal injection of 4% chloral hydrate (10 μl/g), the brain was rapidly removed and rinsed briefly in ddH$_2$O. The tissues were immersed in impregnation solution made by mixing equal volumes of solution A and B, and then stored at RT for 2 weeks in the dark. Tissues were transferred to solution C and store at 4 °C in dark for 7 days. Coronal slices (150 μm thickness) were prepared using a vibrating tissue slicer (Leica VT1000S) in solution C. Slices were mounted on gelatin-coated slides (FD NeuroTechnologies, PO101) with solution C and dry naturally at RT. Slices were rinsed in ddH$_2$O twice and placed in a mixture consisting of 1 part solution D, 1 part solution E and two part ddH$_2$O for 10 min. Slices were rinsed in ddH$_2$O twice, dehydrated in 50%, 75%, 95%, and 100% ethanol for 4 min, cleared with xylene for three times, sealed with permount (Sigma, 06522). Images of hippocampus CA1 pyramidal neurons were captured by an Olympus BX61 microscope with 100× objective lens. The Sholl analysis, spine density and the morphological classification of the spines were calculated by Image J software (*Ferreira et al., 2014*). The morphological classification of spines as previously described (*Okamura et al., 2004*; *Malone et al., 2008*; *Rochefort and Konnerth, 2012*): mushroom, big protrusions with a well-defined neck and a very voluminous head; stubby, short protrusions without a distinguishable neck and head; long-thin, protrusions with a long neck and a clearly visible small bulbous head; filopodia, thin, hair-like protrusions without bulbous head.

## Preparation of subcellular fractionation

Hippocampal tissues were homogenized in buffer C (320 mM sucrose, 20 mM Tris-HCl [pH = 8.0], 2 mM EDTA and 200 μg/ml PMSF) supplemented with protease inhibitors (Merck Chemicals) and centrifuged at 800× g for 10 min at 4 °C to result in P1 and supernatant S1. The supernatant S1 was centrifugated at 12,000× g for 15 min at 4 °C to yield P2 and S2. Then P2 fraction was resuspended in buffer D (20 mM Tris-HCl [pH = 8.0], 1 mM EDTA, 100 mM NaCl, and 1.3% Triton X-100) and centrifuged at 100,000× g for 1 hr to produce a supernatant S3 and a pellet P3. The pellet represented PSD fraction, and the protein concentration was measured using BCA protein assay kit (Thermo, 23225).

## Surface biotinylation assay

Acute mouse coronal slices containing hippocampus region (250 µm) were prepared from anesthetic mice using a vibrating tissue slicer (Leica VT1000S) in ice-cold artificial cerebrospinal fluid (ACSF) containing: 125 mM NaCl, 2.5 mM KCl, 1.25 mM NaH$_2$PO$_4$, 1 mM MgCl$_2$, 2 mM CaCl$_2$, 26 mM NaHCO$_3$ and 25 mM D-glucose, bubbled with 95% O$_2$ / 5% CO$_2$. After recovery for 30 min at 37°C, slices were incubated with 0.5 mg/ml EZ-link-sulfo-NHS-SS-Biotin (Thermo, 21331) in ACSF for 30 min. The biotin solution was then removed, and the remaining unreacted biotin was quenched by the addition of 50 mM Tris solution (pH = 7.4) for 2 min. The slices were then washed with ice-cold PBS and lysed with RIPA buffer supplemented with protease inhibitors. Biotinylated proteins were separated from other proteins using streptavidin agarose beads (Thermo, 20349) overnight at 4°C. The agarose beads were then washed three times with RIPA buffer and 500× g, 2 min centrifuge. The biotinylated proteins were extracted with SDS-PAGE loading buffer and boiled for 3 min before western blot.

## Transmission electron microscopy (TEM)

Mice (P30) were transcranial perfused with saline and ice-cold fixative (4% PFA containing 0.1% glutaraldehyde), brains were removed and stored at 4 °C for 2.5 hr in fixative. Coronal slices containing hippocampus region (250 µm) were cut on a vibratome and kept in ice-cold fixative solution. Small blocks of tissue (1 mm$^2$) from the CA1 region were excised under a microscope. The CA1 tissues were then rinsed for 6 times with 0.1 M PB and post-fixed in 1% OsO$_4$ for 30 min, rinsed for three times with ddH$_2$O and stained with 2% uranyl acetate for 30 min at room temperature. After dehydrating through a graded series of ethanol (50%, 70%, 90%, and 100%), the samples were embedded in an epoxy resin. Ultrathin sections (90 nm) were cut using an ultra-microtome (Leica), stained with lead citrate solution, and mounted on grids. EM images were captured at ×11,000 and ×68,000 magnification using a Tecnai transmission electron microscope (FEI, Hillsboro, OR). CA1 synapses were identified by the presence of a presynaptic bouton with at least two synaptic vesicles within a 50 nm distance from the cellular membrane facing the spine, a visible synaptic cleft, and PSD (*Spano et al., 2019*). Image J software was used to analyze the PSD length and PSD thickness.

## Electrophysiology

We used 1% barbital sodium to anesthetize mice and removed the brain of mice rapidly and placed it in ice-cold, high-sucrose cutting solution containing (in mM): 194 Sucrose, 30 NaCl, 4.5 KCl, 0.2 CaCl$_2$, 2 MgCl$_2$, 1.2 NaH$_2$PO$_4$, 26 NaHCO$_3$, 10 Glucose. We cut slices coronally in the high-sucrose cutting solution with vibratome, and transferred the slices to an incubation chamber with artificial cerebrospinal fluid (ACSF) containing (in mM): 119 NaCl, 2.5 KCl, 2.5 CaCl$_2$, 1.3 MgSO$_4$, 1.0 NaH$_2$PO$_4$, 26.2 NaHCO$_3$, 11 Glucose. Slices were incubated at 34 °C for 30 min and then transferred to 27 °C for at least 1 hr before recording. During recording, the slices were placed in a recording chamber constantly perfused with warmed ACSF (32 °C) and gassed continuously with 95% O$_2$ and 5% CO$_2$. Pipettes for Whole-cell recordings (3–5 MΩ) were filled with a solution containing (in mM): 115 CsMeSO$_4$, 20 CsCl, 10 HEPES, 2.5 MgCl$_2$, 4 Na-ATP, 0.4 Na$_2$-GTP, 10 Na$_2$phosphocreatine, 0.6 EGTA, 5 QX-314 (pH 7.2–7.4, osmolarity 290–310). Data were collected with a MultiClamp 700B amplifier and analyzed by pClamp10 software (Molecular Devices, Sunnyvale, USA). The initial access resistance was <25 MΩ and was monitored throughout each experiment. Data were discarded if the access resistance changed 20% during whole recording. Data were filtered at 2 kHz and digitized at 10 kHz.

For field excitatory postsynaptic potential recording, DG zone was cut down before recording to avoid the generation of epileptic wave. A concentric bipolar stimulating electrode was placed in the stratum radiatum to evoke EPSP and the recording electrode filled with ACSF was placed at 200–400 µm away from the stimulus electrode at the same region. Increasing stimulus intensity step by step to find the maximal response and then the baseline response was adjusted at 1/3 to 1/2 of the maximal response. After the steady baseline response was recorded for 30 min, a strain of 100 Hz-100 pulses high frequency stimulus (1× HFS) or four strains of 100 Hz-100 pulses high frequency stimulus applied with 5 min intervals (4× HFS) was delivered to induce LTP and a train of 2 Hz-900 pulses low-frequency stimulus was delivered to induce LTD. The induced post response was recorded for 1 hr (1× HFS) or 2 hr (4× HFS). The slope within 2ms at initial EPSP was calculated as the value for field EPSP. The magnitude of LTP and LTD was calculated based on the EPSP values 55–60 min (1× HFS) or 130–135 min (4× HFS) after the end of the induction protocol.

For whole-cell LTP recording, 100 µmol/L picrotoxin was added in recycling ACSF to block GABAR-mediated current. The stimulus electrode was placed in the stratum radiatum to evoke EPSC at 0.1 HZ and AMPAR-mediated current was recorded at –70 mV. After the steady baseline response was recorded for 5 min, the cell was held at 0 mV and received two strains of 100 Hz-100 pulses high frequency stimulus with 20 s intervals. And then the cell was held at –70 mV to record for 40 min. Here, the HFS must be given within 10 min of achieving the whole-cell recording to avoid 'wash-out' of LTP. The magnitude of LTP was calculated based on the EPSC values 35–40 min after the end of the induction protocol.

For NMDAR-mediated current and AMPAR-mediated current recording, 100 µmol/L picrotoxin was added in recycling ACSF to block GABAR-mediated current. AMPAR-mediated current was recorded at –70 mV. A mixed current for AMPAR-mediated current and NMDAR-mediated current was recorded at +40 mV. The current at 50ms after stimulus was counted for NMDAR-mediated current. The AMPAR/NMDAR ratio was calculated as the peak of the averaged AMPAR-mediated EPSC (30–40 consecutive events) at –70 mV divided by the averaged NMDAR-mediated EPSC (30–40 consecutive events).

Input-Output curves were established by plotting the EPSC amplitude against various intensities of the test pulse, ranging from 25 µA to 50 µA in 5 µA steps. NMDAR-mediated EPSCs were recorded in the presence of 20 µM CNQX and 100 µM picrotoxin and at +40 mV holding potential. AMPAR-mediated EPSCs were recorded in the presence of 50 µM AP5 and 100 µM picrotoxin and at –70 mV holding potential.

For mini excitatory synaptic transmission recording, 100 µmol/L picrotoxin and 1 µmol/L TTX was added into recycling ACSF to block GABA currents and sodium currents respectively. Recording was conducted at –70 mV. Chemical LTP (cLTP) was induced by treatment of hippocampus slices with glycine (200 µM) to activate NMDA receptors. Slices were treated with glycine in the $Mg^{2+}$ free ACSF for 10 min, then transferred to the recording chamber with ACSF containing 1 µM tetrodotoxin, 100 µM picrotoxin for patch recordings. Mini analysis software was used to count the mini EPSCs.

Paired-pulse facilitation (PPF) was examined by paired stimulations at different intervals.

## Behavioral experiments

All behavioral experiments were performed with the age of 2- to 3-month-old male littermates. All animals were given at least 1 hr of habituation time before testing. KIF2C(WT) re-expression assay for behavioral tests used a doxycycline-induced viral expression system.

## Open-field test

The open-field test was used to evaluate the autonomous behavior, exploratory behavior, and tension of animals in a new environment. Through dynamic monitoring of exploratory behavior and spontaneous activity, it is often used to test basic exercise volume and anxiety level. The mice were put in an area of 40 cm× 40 cm × 40 cm box and exploring freely for 30 min using a video tracking system. Data were analyzed using ANY-maze software.

## Y-maze

The Y-maze was used to evaluate discriminative learning, short-term working memory and reference memory test. It is based on the innate curiosity of the rodents to explore an arm that has not been previously visited. The Y-shaped maze is composed of three equal-length arms (40 cm×9 cm × 16 cm). The angle between each two arms is 120 degrees. The inner arms and the bottom of the maze are painted dark gray. The mice were placed in the center of the maze and allowed to freely explore the three arms for 5 min. Take the animals' four feet into the arm as the standard. One alternation would be recorded if all arms were entered successively at a time. The numbers of arm entries were recorded to evaluate the locomotion of animals in the Y-maze. Spontaneous alternation was calculated as total alternations/the number of maximum alternations (total arm entries-2). Error % was calculated as total errors/the number of maximum errors (total arm entries-2).

## Elevated zero maze

Elevated zero maze was used to investigate the animal's anxiety state which was based on the animal's exploring characteristics of the new environment and the fear of hanging open arms to form conflicting behaviors. The annular platform (60 cm in diameter, 5 cm track width) was evaluated 50 cm above

the floor and divided equally into four quadrants. Two opposite quadrants were 'open' and others were surrounded by 16 cm high blue, opaque walls. The mice were randomly placed in closed quadrants and allowed to freely to explore for a period of 5 min using an overhead video tracking system. Dependent measures were: time in open quadrants, open entrances, number of head dips, number of stretch attendance postures (SAP) and number of four paws in open section.

## Morris water maze

Morris water maze is the most classic behavioral experiment that evaluates animal learning, memory, cognition, and space exploration ability. The water maze consists of a circular pool (radius 120 cm) filed with water containing nontoxic titanium pigment to make the platform invisible. Mice were trained to find the hidden platform (5 cm) and placed gently in different quadrants each time. The maximum searching period allotted was 60 s. If a mouse did not find the platform, it will be placed on the platform by the experimenter and remained for 10 s. Training was performed for five days and tested on the sixth day. For the training sessions, the time taken to find the platform, velocity, and distance traveled were recorded. For the test session, the platform was removed. The time spent in the four quadrants of the pool were recorded.

## Self-grooming

The mice were placed individually in a clean cage with fresh bedding. After 10-min habituation, mice were monitored with a camera and recorded for 10 min to score the spontaneous grooming behavior. Time of grooming was recorded.

## Three-chambered social test

Three-chamber social interaction test was carried out as described previously (*Silverman et al., 2010*). Briefly, two empty steel cylinders were respectively placed in the left and right rectangular. The test mice were individually placed in the center chamber for a 10-min habituation. Then the mice encountered a stranger C57BL /6 N mouse (S1) in the cage for 10 min. The social novelty test was then arranged and another stranger C57BL/6 N (S2) was placed in the cage and let the test mouse explored freely for 10 min. The close interaction time of the test mouse spent in each chamber was recorded. The ratio of (S1-E) to (S1+ E) was calculated as the preference index (S1-E), and the ratio of (S2-S1) to (S2+ S1) was measured as the preference index (S2-S1).

## Cued fear conditioning

Mice from each genotype were placed to the conditioning chamber for a 160-s habituation and then received three conditional stimulus (CS)- unconditional stimulus (US) trails inter-spaced by 40 s inter-trial intervals. They were given 60 s stimulus-free interval before returned to their home cage. The conditional stimulus comprised a 20 s discrete tone (80 dB, 2.8 kHz), which was followed by an unconditional stimulus at the last 2 s (US, 0.55 mA foot shock). To examine the condition response to the background contextual cues, the mice were placed back in the conditioning chamber for 6 min in the absence of the CS and US stimuli after training 24 hr, 7d. To examine the condition response to the CS, the mice were placed in a distinct and modified context and given a 6 min CS tone after training 48 hr, 8d. The context maintenance index is the ratio of (24hr-7d) to 24 hr freezing time, and the CS maintenance index is the ratio of (48hr-8d) to 48 hr freezing time. For all procedures, the animal freezing behaviors were monitored using a manufacturer's software.

## Prepulse inhibition (PPI) test

PPI was performed as described previously (*Maekawa et al., 2010*). Briefly, in PPI test, prepulses (PP) were set at 70, 74, 78, and 82 dB, and pulse (P) was set at 120 dB. Percentage PPI was calculated as [(P amplitude – PP amplitude)/ P amplitude] × 100%.

## Statistics

Data were analyzed using GraphPad Prism 8.0 (GraphPad Software, San Diego, CA), Excel 2017 (Microsoft, Seattle, WA), and Sigmaplot 12.0 (Systat Software, Erkrath, Germany). Data analyses were blind to experimental conditions until the data were integrated. Standard deviations for controls were calculated from the average of all control data. Statistical difference was determined using a Student's

*t* test or one-way ANOVA with Tukey's *post hoc* tests or two-way ANOVA with Bonferroni correction. The accepted level of significance was $p < 0.05$. *n* represents the number of preparations, experimental repeats or cells as indicated in figure legends. Data are presented as mean ± SEM.

## Acknowledgements

We are grateful to research assistant Shuang-shuang Liu from the Core Facilities of Zhejiang University School of Medicine, Dr. San-hua Fang and research assistant Dao-hui Zhang from the Core Facilities of Zhejiang University Institute of Neuroscience, and research assistant Bei-bei Wang from the center of Cryo-Electron Microscopy of Zhejiang University for the operation of the instruments. We are grateful to Dr. Xiao-dong Wang and Xiao-Jun Wang from School of Brain Science and Brain Medicine of Zhejiang University for the guidance on statistical methods.

## Additional information

### Funding

| Funder | Grant reference number | Author |
| --- | --- | --- |
| National Natural Science Foundation of China | 31970902 | Junyu Xu |
| National Natural Science Foundation of China | 3192010300 | Jianhong Luo |
| National Natural Science Foundation of China | 32000692 | Xintai Wang |
| National Natural Science Foundation of China | 31871418 | Jianhong Luo |
| National Natural Science Foundation of China | 81821091 | Jianhong Luo |
| Natural Science Foundation of Zhejiang Province | LD19H090002 | Jianhong Luo |
| Natural Science Foundation of Zhejiang Province | LR19H090001 | Junyu Xu |
| Special Project for Research and Development in Key areas of Guangdong Province | 2019B030335001 | Junyu Xu |
| Research Grants Council of the Hong Kong SAR, China | 16103520 | Jun Xia |

The funders had no role in study design, data collection and interpretation, or the decision to submit the work for publication.

### Author contributions

Rui Zheng, Conceptualization, Data curation, Formal analysis, Investigation, Methodology, Software, Validation, Writing – original draft; Yonglan Du, Data curation, Formal analysis, Investigation, Methodology, Software, Validation; Xintai Wang, Formal analysis, Funding acquisition, Methodology, Software, Validation; Tailin Liao, Formal analysis, Methodology, Validation; Zhe Zhang, Formal analysis, Validation; Na Wang, Conceptualization, Methodology; Xiumao Li, Methodology, Validation; Ying Shen, Supervision, Validation, Writing – review and editing; Lei Shi, Validation, Visualization, Writing – review and editing; Jianhong Luo, Funding acquisition, Validation, Writing – review and editing; Jun Xia, Validation, Writing – review and editing; Ziyi Wang, Data curation, Formal analysis, Methodology, Resources, Supervision, Validation, Writing – review and editing; Junyu Xu, Conceptualization, Data curation, Formal analysis, Funding acquisition, Project administration, Resources, Software, Supervision, Validation, Writing – original draft, Writing – review and editing

## Author ORCIDs

Na Wang http://orcid.org/0000-0002-1438-1508
Ying Shen http://orcid.org/0000-0001-7034-5328
Lei Shi http://orcid.org/0000-0001-8695-3432
Jun Xia http://orcid.org/0000-0001-9308-2188
Junyu Xu http://orcid.org/0000-0002-1911-3553

## Ethics

All animal experiments were performed in accordance with the ethical guidelines of the Zhejiang University Animal Advisory Committee (2020-19500#), and were in complete compliance with the National Institutes of Health Guide for the Care and Use of Laboratory Animals. All surgeries were performed under sodium pentobarbital anesthesia, and every effort was made to minimize suffering.

## Decision letter and Author response

Decision letter https://doi.org/10.7554/eLife.72483.sa1
Author response https://doi.org/10.7554/eLife.72483.sa2

## Additional files

### Supplementary files

- Transparent reporting form
- Source data 1. All original files of blots and gels with the relevant bands.

### Data availability

We have uploaded the source data files together with manuscript.

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

# Appendix 1

**Appendix 1—key resources table**

| Reagent type (species) or resource | Designation | Source or reference | Identifiers | Additional information |
|---|---|---|---|---|
| Strain, strain background (*Mus musculus*, male/female) | C57BL/6 | Shanghai SLAC Laboratory Animal C.,Ltd | | |
| Strain, strain background (*Mus musculus*, male/female) | *kif2c*<sup>flox/flox</sup> mouse | Model Animal Research Center of Nanjing University | | |
| Strain, strain background (*Mus musculus*, male/female) | *Nestin*<sup>Cre</sup> mouse | Zhejiang University | *Wang et al., 2019* | |
| Cell line (*Homo-sapiens*) | HEK 293T | ATCC | CRL-3216 | |
| Genetic reagent (virus) | pAKD-CMV-bGlobin-mcherry-H1-shRNA | ObioTechnology | | |
| Genetic reagent (virus) | pAAV-CMV-KIF2C-3FLAG-P2A-mNeonGreen-CW3SL | ObioTechnology | | |
| Genetic reagent (virus) | pAAV-CMV-KIF2C(G491A)–3FLAG-P2A-mNeonGreen-CW3SL | ObioTechnology | | |
| Genetic reagent (virus) | rAAV-TRE3G- KIF2C-3*Flag –2A-mCherry-WPRE-pA | BrainVTA | | |
| Antibody | Anti-KIF2C, (rabbit polyclonal) | Proteintech | Cat# 12139–1-AP, RRID:AB_2877829 | IF (1:500), WB (1:500) |
| Antibody | Anti-PSD-95, (mouse monoclonal) | Abcam | Cat# ab2723, RRID:AB_303248 | IF (1:1000) WB (1:1000) |
| Antibody | Anti-MAP2, (chicken polyclonal) | Abcam | Cat# ab5392, RRID:AB_2138153 | IF (1:2000) |
| Antibody | anti-GFP (rabbit polyclonal) | Abcam | Cat# ab6556, RRID:AB_305564 | IF (1:2000) |
| Antibody | Anti-α-tubulin, (rabbit monoclonal) | Abcam | Cat# ab52866, RRID:AB_869989 | IF (1:1000) |
| Antibody | Anti-mCherry, (rabbit polyclonal) | Abcam | Cat# ab167453, RRID:AB_2571870 | WB (1:2000) |
| Antibody | Anti-EB3, (rabbit monoclonal) | Abcam | Cat# ab157217, RRID:AB_2890656 | WB (1:10,000) |
| Antibody | Anti-α-tubulin, tyrosinated, (rat monoclonal) | Millipore | Cat# MAB1864-I, RRID:AB_2890657 | IF (1:1000) |
| Antibody | Anti-synaptophysin, (mouse monoclonal) | Millipore | Cat# MAB368, RRID:AB_94947 | IF (1:1000) WB (1:2000) |
| Antibody | Anti-GAPDH, (mouse monoclonal) | Millipore | Cat# MAB374, RRID:AB_2107445 | WB (1:10,000) |

*Appendix 1 Continued on next page*

*Appendix 1 Continued*

| Reagent type (species) or resource | Designation | Source or reference | Identifiers | Additional information |
|---|---|---|---|---|
| Antibody | Anti-GluA1, (rabbit monoclonal) | Millipore | Cat# 04–855, RRID:AB_1977216 | WB (1:1000) |
| Antibody | Anti-GluA2, (mouse monoclonal) | Millipore | Cat# MAB397, RRID:AB_2113875 | WB (1:2000) |
| Antibody | Anti-GluN2A, (rabbit polyclonal) | Cell Signaling Technology | Cat# 4205, RRID:AB_2112295 | WB (1:2000) |
| Antibody | Anti- GluN2B, (rabbit polyclonal) | Cell Signaling Technology | Cat# 4207, RRID:AB_1264223 | WB (1:2000) |
| Antibody | Anti-GluN1, (rabbit polyclonal) | Cell Signaling Technology | Cat# 5704, RRID:AB_1904067 | WB (1:2000) |
| Antibody | Anti-flotillin, (mouse monoclonal) | BD Bioscience | Cat# 610820, RRID:AB_398139 | WB (1:1000) |
| Antibody | Anti-β-tubulin, a (mouse monoclonal) | Santa Cruz | Cat# sc-5274, RRID:AB_2288090 | WB (1:2000) |
| Antibody | Anti- bassoon, (mouse monoclonal) | Enzo Life Sciences | Cat# ADI-VAM-PS003-F, RRID:AB_11181058 | IF (1:1000) |
| Antibody | Anti- N-cadherin, (rabbit polyclonal) | Bioworld Technology | Cat# BS2224, RRID:AB_1664028 | WB (1:1000) |
| Chemical compound, drug | CNQX | Sigma-Aldrich | Cat# C239; CAS: 115066-14-3 | 20 μM |
| Chemical compound, drug | AP-V | Sigma-Aldrich | Cat# A8054; CAS: 79055-68-8 | 50 μM |
| Chemical compound, drug | Doxycycline Hyclate | Selleck | Cat# S4163; CAS:24390-14-5 | 40 μg/ml |
| Chemical compound, drug | Glycine | Sigma-Aldrich | Cat# G8898; CAS:56-40-6 | 200 μM |
| Chemical compound, drug | bicuculline | Abcam | Cat# ab120108; CAS: 40709-69-1 | 20 μM |
| Chemical compound, drug | Picrotoxin | Sigma-Aldrich | Cat# P1675; CAS: 124-87-8 | 100 μM |
| Commercial assay or kit | SYBR Green qPCR Master Mix | QIAGEN | Cat# 204,057 | |
| Commercial assay or kit | Nissl staining Kit | Beyotime | Cat# C0117 | |
| Commercial assay or kit | FD Rapid GolgiStain Kit | FD NeuroTechnologies | Cat# PK401 | |
| Commercial assay or kit | BCA Protein Assay Kit | Thermo Fisher | Cat# 23,227 | |
| Software, algorithm | GraphPad Prism (8.0) | GraphPad Software | https://www.graphpad.com/scientific-software/prism/ | RRID: SCR_002798 |
| Software, algorithm | ImageJ | National Institutes of Health | https://imagej.nih.gov/ij/ | RRID:SCR_003070 |
| Software, algorithm | ANY-maze tracking software | Stoelting Co | https://www.stoeltingco.com/anymaze.html | RRID:SCR_014289 |
| Software, algorithm | PlusTipTracker | UTSouthwestern Medical Center Danuser Lab | https://www.utsouthwestern.edu/labs/danuser/software/ | |

*Appendix 1 Continued on next page*

*Appendix 1 Continued*

| Reagent type (species) or resource | Designation | Source or reference | Identifiers | Additional information |
|---|---|---|---|---|
| Software, algorithm | Sigmaplot | systatsoftware | http://www.systatsoftware.cn/ | RRID:SCR_003210 |
| Software, algorithm | MATLAB R2017b | MathWorks | https://se.mathworks.com/products/matlab.html | RRID: SCR_001622 |
| Sequence-based reagent | *KIF2C* floxP _F | This paper | PCR primers | GGT CCA GCT CTT TAC TGA TGT GTT C |
| Sequence-based reagent | *KIF2C* floxP _R | This paper | PCR primers | ACA AAG CAA GTC CAG GTC CAA G |
| Sequence-based reagent | *Nestin*-Cre_F | This paper | PCR primers | TGC AAC GAG TGA TGA GGT TC |
| Sequence-based reagent | *Nestin*-Cre_R | This paper | PCR primers | GCT TGC ATG ATC TCC GGT AT |
| Sequence-based reagent | *kif2c*_F | This paper | PCR primers | TGC CGT TGT TGA TGG TCA GTG |
| Sequence-based reagent | *kif2c*_R | This paper | PCR primers | GGA GAC ACT TGC TGG GAA CAG |
| Sequence-based reagent | *shkif2c*_F | This paper | PCR primers | TGG ATC GAA GGA GGT ACC AC |
| Sequence-based reagent | *shkif2c*_R | This paper | PCR primers | CAC TGA CCA TCA ACA ACG GCA |
| Sequence-based reagent | *gapdh*_F | This paper | PCR primers | AGG TCG GTG TGA ACG GAT TTG |
| Sequence-based reagent | *gapdh*_R | This paper | PCR primers | TGT AGA CCA TGT AGT TGA GGT CA |
| Peptide, recombinant protein | BDNF | Sigma-Aldrich | Cat# GF301 | 50 ng/mL |

