## [Editor Report]

In this manuscript, the authors report a set of exciting and novel data suggesting a critical role of the microtubule depolymerizing kinesin-13 KIF2C/MCAK in glutamate receptor trafficking, synaptic plasticity and behaviours related to learning and memory. The authors use multiple experimental approaches, ranging from electron microscopy to mouse behavioral analysis to support the conclusions, which are convincing and provide novel insights into the in vivo functions of KIF2C in the regulation of excitatory synaptic structure and function and cognitive behaviors.

---

## [Decision Letter]

**Decision letter after peer review:**

Thank you for submitting your article "KIF2C regulates synaptic plasticity and cognition by mediating dynamic microtubule invasion of dendritic spines" for consideration by *eLife*. Your article has been reviewed by 3 peer reviewers, one of whom is a member of our Board of Reviewing Editors, and the evaluation has been overseen by Anna Akhmanova as the Senior Editor. The following individual involved in review of your submission has agreed to reveal their identity: Yu-Tian Wang (Reviewer #2).

Summary

The authors have provided data for the function of the microtubule depolymerizing kinesin, KIF2C/MCAK, in hippocampal neuron function, synaptic transmission, plasticity and mouse behavior. As such, many different techniques are used to determine the function of this protein. Previous work has implicated a similar family member, KIF2A, in axonal branching, however KIF2C appears to be functioning primarily in dendrites. This is an intriguing hypothesis and would provide important data for understanding the function of a microtubule depolymerase in nervous system function. Although copious amounts of data are shown and the manuscript is relatively clearly written, the authors oftentimes overinterpret the data and do not use the appropriate statistical tests and comparisons. The manuscript title and conclusions made in the Abstract and Discussion suggesting that the mechanism of KIF2C action is due to changes in microtubule invasion of dendritic spines is not substantiated. Because KIF2C conditional knock out neurons clearly have defects in microtubule dynamics throughout the dendritic shafts, many of the defects ascribed to microtubule invasion of spines are potentially indirect and may not be responsible for changes in synaptic transmission or behavior. More careful and appropriate analysis of their current data and additional experiments, specifically regarding the possibility of KIF2C leaving spines during synaptic activity and the dynamics of MT invasion of spines, would be needed to determine the mechanism of KIF2C action in dendrites. Thus, although interesting, the manuscript requires substantial revisions.

Essential revisions:

1. An important part of the proposed mechanism of KIF2C action relies on the data in 2H and I. The authors argue that KIF2C is leaving dendritic spines and concentrating in the dendritic shaft after cLTP treatment. However, only colocalization of PSD95 and KIF2C is quantified, and this quantification is a percentage. It is not clear to what the percentage refers – the % of spines where they are colocalized? How is colocalization determined? What is the cutoff? Even if all of these measures were indicated, much more rigorous quantification would have to be done to provide convincing evidence that KIF2C is leaving spines and concentrating in the dendrite shaft after cLTP (as the show in their model in Figure 8). Specifically, both fixed cell imaging of endogenous KIF2C in a much more rigorous fashion, taking into account spine volume changes and comparison with another protein known to leave spines after cLTP. Imaging GFP-KIF2C in individual spines before and after cLTP and showing a subsequent decrease in the spine and increase in the shaft would also provide more compelling data.

2. The authors overinterpret the data from Figure 6C. In this experiment they treated cells with BDNF, fixed the neurons and labeled them with an antibody to dynamic MTs (tyrosinated tubulin). They show that the percent of spines containing a MT increased with BDNF in WT neurons but not in controls. They also show that cKO neurons have slightly more spines containing MTs. This methodology is useful, in addition to live-cell imaging of EB3 entries into spines, but it is difficult to quantify MT invasions because tyrosinated tubulin is also cytoplasmic and spines can change sizes with treatment. Thus, it is difficult to determine which spines contain a MT vs. just increased volume of tyrosinated tubulin. Without live cell imaging of labeled EB3 or tubulin these data are not sufficient to make claims about MT invasion. Also, only live cell imaging would be able to differentiate increased invasion of MTs vs. increased retention of MTs in spines. It is unclear what is happening in the KIF2C cKO neurons. Thus, they can't claim in the discussion that "KIF2C deficiency also increased the frequency of MT invasion under basal conditions" (line 318).

3. In addition to the points made in #2 the fact that MT dynamics are affected in the dendrites suggests that changes in MT invasion of spines are indirect. KIF2C is clearly affecting MT dynamics throughout the neuron. Thus, one would expect changes in MT invasion of spines. Therefore, changes in synaptic transmission may have nothing to do with MT invasion of spines but rather they are due to changes in MT-based transport throughout the dendritic arbor.

Non-essential comments:

1. A major effect of KIF2C seems to be constraining the number of AMPARs in the postsynaptic membrane. However, its underlying mechanism remains uncharacterized – is it a result of inhibiting trafficking of the receptor to the synaptic site, its exocytosis into the plasma membrane, or destabilizing the receptor on the postsynaptic membrane? Can the authors provide any clarification on this?

2. Figure 2E：Is cLTP-induced increase in the spine width associated with an increase in the synaptic AMPARs and amplitude of mEPSCs? If so, would they also be blocked by shRNA?

3. Figure 3: N/A ratio changes may not indicate an increase in AMPAR-mediated synaptic transmission as it is not clear if NMDAR-mediated mEPSCs change or not. The authors should try to evaluate the changes in synaptic transmission by comparing the input-output curves between control and shRNA-treated neurons.

4. Figure 4: again, there should be a detailed analysis of the input-output curves in slices of WT and cKO mice. The individual traces on the left in 4E are not representative as the reduction of N/A ratio in cKO was not as obvious as shown in the bar graph on the right, and need to be changed. The impairment in LTP induced by a single HFS may not be sufficient. In order to further substantiate the impaired LTP, the authors should also attempt to maximize the LTP induced by 4xHFS.

5. Hippocampal re-expression of KIF2C rescues mEPSC, LTP, and behavioral phenotypes. These are nice sets of data suggesting KIF2C directly regulates the synaptic and behavioral phenotypes. However, I wonder whether the same KIF2C re-expression rescues baseline and BDNF-dependent MT invasion, which is an essential part of the conclusions together with synaptic phenotypes.

6. Biochemical work and ICC of hippocampal neurons suggests KIF2C is present postsynaptically, but it also colocalizes with Bassoon (Figure 1E). However, the authors discount the colocalization with a presynaptic label and only focus on the postsynaptic localization. A line scan is used to show colocalization, but this not sufficient. Colocalization analysis and Pearson's correlation coefficients should be computed to definitively show there is more colocalization with PSD95 than Bassoon.

7. Throughout Figure 6 and Figure 7 the authors use a KIF2C point mutant (G491A) that has been shown previously to have little MT depolymerization activity. However, they do not directly compare the point mutant to the WT KIF2C when they attempt to rescue the synaptic transmission phenotypes in the cKO mouse. Instead they compare each to a control and use a Student's t-test to indicate that the cKO + WT protein is significantly different than control, while the cKO + G491A is not. They need to compare cKO + WT directly to G491A and controls via ANOVA and show these two "rescues" are different. Moreover, addition of a WT control would be needed if they wanted to claim that the cKO + WT is the same as WT control (i.e. a full rescue).

8. In Figure 2 and 3 it is not clear why immunoblots are not used to show decreases in protein expression after knockdown. Only mRNA quantification is shown.

9. In Figure 2 KIF2C knockdown seems to have no effect on basal levels of spine density or spine head width. It is only after cLTP that significant differences are noted. However, in Figure 1D a Student's t-test is used, while in Figure 1E an ANOVA is used. Both of these graphs, as well as Figure 1G, should be analyzed with an ANOVA because more than two conditions are being compared. The minor differences (p<0.05) in Figure 1D and 1G are likely to be non-significant when analyzed with an ANOVA.

10. Bar graphs and scatter plots are used randomly throughout the paper. Scatter plots, or better yet SuperPlots (Lord et al., 2020, JCB, 219(6):e202001064), should be used throughout the paper.

11. The data in Figure 5E and F seem to be inconsistent. Both NMDAR and AMPAR appear to increase, although AMPAR to a greater extent than NMDARs, but the N/A ratio is significantly less in the cKO. Furthermore, only a p-value is provided. No statistical tests are mentioned.

12. In Figure 6B it is unclear why measurements of EB3 speed manually and using Tip Tracker software would result in such different growth speeds (EB3 speed vs. MT growth speed). The MT growth speeds are double the EB3 speeds. Shouldn't these be the same speeds?

13. In Figure 7C (lines 260-261) the authors claim the maintenance of fear memory is impaired in KIF2C cKO mice, however it appears that the maintenance is preserved – the two lines are parallel. Rather the fear memory is decreased throughout the periods of measurement. Also, these measures need to be analyzed with a repeated measures ANOVA, rather than a t-test.

14. Lines 320-21: The authors suggest that MTs bring AMPAR to the synapse. To my knowledge, this has not been shown here or elsewhere in the literature.

15. The model in Figure 8 is inaccurate. The authors want to show increased invasion of MTs into spines but show multiple simultaneous invasions. They never image MT dynamics in spines to show this.

16. The authors need to carefully review the writing contained within the manuscript, including the figure legends. The corresponding legend for Supplementary Information Figure S1 needs to be updated to reflect A-D as opposed to an abridged version A-B. Figure 4A includes "sgRNA" as opposed to "shRNA". Supplementary Information Figure S4 fails to mention B' in the legend and currently states "Number of strenching" as the title for the y-axis. There are also a number of confusing/distracting sentences within the body of the manuscript that need remedying.

17. In the description of Figure 2 (line 111), the authors state that spine morphology was examined at DIV 16-18 when the methods and figure legend state that this assessment occurred at DIV 18-20.

18. Line 135: Figure S1 is not referenced regarding the cultured neurons.

*Reviewer #1:*

This is a careful and thorough in vivo characterization of the synaptic function of the KIF2C protein using biochemical, cell biological, electrophysiological, mouse genetic, and behavioral analyses. The conclusions suggest that KIF2C regulates activity-dependent MT invasion, spine morphology, synaptic transmission, and synaptic plasticity. The rescue of synaptic and behavioral phenotypes by the use of KIF2C re-expression in the hippocampus is a key component of the manuscript. The overall conclusions are largely convincing and provide novel insights into the in vivo functions of KIF2C in the regulation of excitatory synaptic structure and function and cognitive behaviors.

*Reviewer #2:*

In this manuscript by Zhang et al., the authors report a set of exciting and novel data suggesting a critical role of KIF2C in AMPAR trafficking, synaptic plasticity and learning/memory related behaviours. This is a comprehensive study with clear logic and ample supporting data. Their results are of high quality and convincing.

*Reviewer #3:*

This manuscript is focused on how the kinesin protein KIF2C, a known microtubule depolymerizing protein, functions in hippocampal dendritic spine plasticity, cognition and behavior. The authors use many approaches, from electron microscopy to mouse behavioral analysis, to determine how KIF2C affects hippocampal neuron function. They suggest KIF2C plays a role in microtubule invasion of dendritic spines, which results in dendritic spine defects, deficits in synaptic transmission and behavior. Although interesting, some results are overinterpreted and not all conclusions are directly supported by the data that is presented.

The authors have provided data for the function of the microtubule depolymerizing kinesin, KIF2C/MCAK, in hippocampal neuron function, synaptic transmission, plasticity and mouse behavior. As such, many different techniques are used to determine the function of this protein. Previous work has implicated a similar family member, KIF2A, in axonal branching, however KIF2C appears to be functioning primarily in dendrites. This is an intriguing hypothesis and would provide important data for understanding the function of a microtubule depolymerase in nervous system function. Although copious amounts of data are shown and the manuscript is relatively clearly written, the authors oftentimes overinterpret the data and do not use the appropriate statistical tests and comparisons. The manuscript title and conclusions made in the Abstract and Discussion suggesting that the mechanism of KIF2C action is due to changes in microtubule invasion of dendritic spines is not substantiated. Because KIF2C conditional knock out neurons clearly have defects in microtubule dynamics throughout the dendritic shafts, many of the defects ascribed to microtubule invasion of spines are potentially indirect and may not be responsible for changes in synaptic transmission or behavior. More careful and appropriate analysis of their current data and additional experiments, specifically regarding the possibility of KIF2C leaving spines during synaptic activity and the dynamics of MT invasion of spines, would be needed to determine the mechanism of KIF2C action in dendrites.

[Editors' note: further revisions were suggested prior to acceptance, as described below.]

Thank you for resubmitting your work entitled "KIF2C regulates synaptic plasticity and cognition in mice through dynamic microtubule depolymerization" for further consideration by *eLife*. Your revised article has been reviewed by 3 peer reviewers and the evaluation has been overseen by Anna Akhmanova as the Senior Editor.

The manuscript has been improved but there are some remaining issues that need to be addressed, as outlined below:

Specifically, the authors need to provide details on the statistical methods used through the manuscript and the reasons why the authors could not perform the suggested live imaging experiments (see below for details).

*Reviewer #1:*

The authors have fully addressed my review comments. I do not have any remaining concerns.

*Reviewer #2:*

The authors have adequately addressed my concerns, and the manuscript can now be recommend for publication in the journal.

*Reviewer #3:*

The authors have taken the comments of the reviewers seriously and have responded with manuscript changes, additional experiments and moderated their interpretations of the data they present. These changes have strengthened the manuscript considerably. However, there are still a few points that the authors did not directly address.

In their response to Essential Revision #1 the authors quantified the distribution of KIF2C with regard to PSD-95 in different ways but the data in Figure 2H and I are still not very convincing. They didn't seem to make any measurements in the dendrite shaft to show that KIF2C was increasing in the shaft while decreasing in the spine. From these data it seems they can just conclude that KIF2C is not colocalizing as much with PSD-95 after cLTP. Additionally, they state that they tried overexpressing KIF2C but this caused damage to the neurons. However, in Figure 6E they show they expressed GFP-KIF2C in cKO neurons. Why could they not image these neurons live and quantify changes in KIF2C in the spine and shaft as requested?

The authors still used a Student t-test for Figure 2G and Figures 6F and G instead of ANOVA. Moreover, although they use ANOVA more appropriately in the revision, they have not indicated what post-tests they have run (Bonferroni, Tukey, etc.?) in their ANOVA analyses throughout the manuscript. These post-hoc tests should be included in the Methods and in the appropriate figure legends.

---

## [Author Response]

Essential revisions:1. An important part of the proposed mechanism of KIF2C action relies on the data in 2H and I. The authors argue that KIF2C is leaving dendritic spines and concentrating in the dendritic shaft after cLTP treatment. However, only colocalization of PSD95 and KIF2C is quantified, and this quantification is a percentage. It is not clear to what the percentage refers – the % of spines where they are colocalized? How is colocalization determined? What is the cutoff? Even if all of these measures were indicated, much more rigorous quantification would have to be done to provide convincing evidence that KIF2C is leaving spines and concentrating in the dendrite shaft after cLTP (as the show in their model in Figure 8). Specifically, both fixed cell imaging of endogenous KIF2C in a much more rigorous fashion, taking into account spine volume changes and comparison with another protein known to leave spines after cLTP. Imaging GFP-KIF2C in individual spines before and after cLTP and showing a subsequent decrease in the spine and increase in the shaft would also provide more compelling data.

We thank the reviewer for pointing this out. The colocalization refers to the percentage number of PSD-95-colocalized KIF2C puncta relative to total number of KIF2C puncta in dendritic protrusions. We totally agree with the reviewer that the increased PSD-95 puncta volume after cLTP would affect the result interpretating. We therefore dug into our data more carefully for more information and made more quantification. We have found that both the density or intensity (i.e., expression level/spine) of synaptic KIF2C was significantly reduced, percentage number of PSD-95-colocalized KIF2C puncta relative to total number of KIF2C puncta was reduced as well. Though the spines enlarged after cLTP, instead of increasing or unchanging, the percentage area of PSD-95-colocalized KIF2C relative to total KIF2C was decreased significantly, indicating that the spine volume change after cLTP was not the disturbing factor in our quantification. Taken together, we are confident that the imaging data did show the translocation of KIF2C out from spines after cLTP. The new data have been incorporated into Figure 2I together with corresponding descriptions in the main text (line 133-135).

The detailed quantification procedures have been incorporated into the Method section. Generally, fluorescence intensity was measured using region measurements function in MetaMorph. Single dendrites were selected randomly. Fixed threshold values were used to optimize highlighting of puncta in green (PSD-95) and red (KIF2C) channels and then applied to all images from the same experiment. For the colocalization quantification, the percentage area of KIF2C overlapping PSD-95 was performed using Measurement Colocalization function. For the quantification of synaptic KIF2C density and intensity, all the puncta on the green (PSD-95) channel were selected and used as a mask for red channel (KIF2C) overlapping to filter out the PSD-95-colocalized KIF2C puncta automatically, using the Process function, and Integrated Morphometric Analysis function was used to identify the puncta and measure their properties.

We have also overexpressed KIF2C in hippocampal neurons and checked its synaptic localization before or after cLTP treatment, according to the reviewer’s suggestion. However, we found that KIF2C overexpression impaired the postsynaptic structure largely, and neurons did not respond to cLTP stimulation. In fact, we found that KIF2C overexpression in WT neurons did more damage than in cKO neurons, as it could not be expressed for too long time in vitro or in vivo. Perhaps overloading of KIF2C in neurons would lead to an overactivated microtubule depolymerization. We suspect that KIF2C's expression in neuron needs to be in a precise balance in order to maintain normal physiological activity, with a mechanism that we have not understood yet.

2. The authors overinterpret the data from Figure 6C. In this experiment they treated cells with BDNF, fixed the neurons and labeled them with an antibody to dynamic MTs (tyrosinated tubulin). They show that the percent of spines containing a MT increased with BDNF in WT neurons but not in controls. They also show that cKO neurons have slightly more spines containing MTs. This methodology is useful, in addition to live-cell imaging of EB3 entries into spines, but it is difficult to quantify MT invasions because tyrosinated tubulin is also cytoplasmic and spines can change sizes with treatment. Thus, it is difficult to determine which spines contain a MT vs. just increased volume of tyrosinated tubulin. Without live cell imaging of labeled EB3 or tubulin these data are not sufficient to make claims about MT invasion. Also, only live cell imaging would be able to differentiate increased invasion of MTs vs. increased retention of MTs in spines. It is unclear what is happening in the KIF2C cKO neurons. Thus, they can't claim in the discussion that "KIF2C deficiency also increased the frequency of MT invasion under basal conditions" (line 318).

We thank the reviewer for the thoughtful analysis and for the suggestion. Our previous result did not provide direct evidence that KIF2C affects MT spine invasion. To just this question, we performed the live-imaging of EB3 and quantified the MT spine invasion events. Our result showed that without KIF2C, the MT invasion events in average each spine increased about 2 folds in an hour. We also provided a representative Kymograph and time-frame images showing a single MT invasion event. Therefore, we feel confident that KIF2C did affect MT invasion under basal condition. The new result has now been incorporated into Figure 6C together with corresponding descriptions in the main text (line 237-238).

3. In addition to the points made in #2 the fact that MT dynamics are affected in the dendrites suggests that changes in MT invasion of spines are indirect. KIF2C is clearly affecting MT dynamics throughout the neuron. Thus, one would expect changes in MT invasion of spines. Therefore, changes in synaptic transmission may have nothing to do with MT invasion of spines but rather they are due to changes in MT-based transport throughout the dendritic arbor.

We thank the reviewer for pointing this out. As our response for #2, we have seen the change in MT invasion by live imaging in cKO mice. We also agree with the reviewer that even though MT invasion of spines is affected, the defects in dendrites also count. Currently, we don’t have a clean and clear method to distinguish KIF2C’s function in dendrites or in spines. KIF2C’s translocation after cLTP induction would be a touch to its regulatory mechanism in spines, but currently we are unable to strip it out from dendritic factor. In order to avoid overinterpretation of the results, we have revised the inaccurate deduction in the article (line 39-42, 213-214, 265-267) as well as in the title, and made a discussion about this issue in the main text (line 335-338, 358-366, 367-388).

Non-essential comments:1. A major effect of KIF2C seems to be constraining the number of AMPARs in the postsynaptic membrane. However, its underlying mechanism remains uncharacterized – is it a result of inhibiting trafficking of the receptor to the synaptic site, its exocytosis into the plasma membrane, or destabilizing the receptor on the postsynaptic membrane? Can the authors provide any clarification on this?

It is interesting to find out how KIF2C regulates surface expression of AMPA receptors. Our results suggest that KIF2C regulates microtubule depolymerization and stability in hippocampal neurons (Figure 6A). Microtubule-dependent trafficking has been typically associated with long-range transport along the dendrites, supporting the continuous supply and redistribution of AMPARs among synapses (1-4) (5, 6), which can also be regulated by neuronal activity (1, 7, 8). There has also been report showing microtubule dynamic stability regulates AMPARs postsynaptic localization and synaptic plasticity(*9*). We therefore speculate that KIF2C constrain AMPARs trafficking to the synaptic sites by microtubule depolymerization and stability. However, other possibilities are not excluded. This could be a very interesting topic for our follow up study.

2. Figure 2E：Is cLTP-induced increase in the spine width associated with an increase in the synaptic AMPARs and amplitude of mEPSCs? If so, would they also be blocked by shRNA?

It has been shown that cLTP induction is characterized by increases in the frequency and amplitude of spontaneous mEPSCs and elevation of CREB phosphorylation (10), altering the size and/or shape of the specialized postsynaptic compartment of the dendritic spine(11-13). Synaptic AMPARs recruitment is required for cLTP-induced spine expansion (13). Our data showed that knockdown of KIF2C blocked the cLTP-induced increase of dendritic spine width (Figure 2E), suggestion a change in synaptic AMPARs and mEPSC amplitude. To direct answer this question, we examined the mEPSCs before and after cLTP in KIF2C knockdown neurons, and found that shKIF2C blocked the cLTP-induced raise of synaptic AMPARs and mEPSCs amplitude. The new results were added in our revised manuscript (Figure 2—figure supplement 1E, F) (line 148-151).

3. Figure 3: N/A ratio changes may not indicate an increase in AMPAR-mediated synaptic transmission as it is not clear if NMDAR-mediated mEPSCs change or not. The authors should try to evaluate the changes in synaptic transmission by comparing the input-output curves between control and shRNA-treated neurons.

We thank the reviewer for the suggestion. In the revised manuscript, we examined the input-output curve of AMPA/NMDA receptor-mediated EPSCs in control and shKIF2C-treated neurons. We found that KIF2C knockdown alters the input-output curves of AMPA receptor but not that of NMDA receptor, indicating that KIF2C shRNA-caused EPSC amplitude alteration is AMPA receptor-mediated. The new result has now been incorporated into Figure 3D-E together with corresponding descriptions in the main text (line 159-163).

4. Figure 4: again, there should be a detailed analysis of the input-output curves in slices of WT and cKO mice. The individual traces on the left in 4E are not representative as the reduction of N/A ratio in cKO was not as obvious as shown in the bar graph on the right, and need to be changed. The impairment in LTP induced by a single HFS may not be sufficient. In order to further substantiate the impaired LTP, the authors should also attempt to maximize the LTP induced by 4xHFS.

We thank the reviewer for the suggestion. We examined the input-output curve of AMPA/NMDA receptor-mediated EPSCs in slices of WT and KIF2C cKO mice. Consistent with the shRNA knockdown result, KIF2C knockout alters the input-output curves of AMPA receptor but not that of NMDA receptor, indicating that KIF2C deficiency-caused EPSC amplitude alteration is AMPA receptor-mediated. The new result has now been incorporated into Figure 5D-E together with corresponding descriptions in the main text (line 200-202).

We suppose that the individual traces the review mentioned is on the left in Figure 5E instead of 4E. They have now been replaced by new ones (Figure 5C in the revised manuscript).

Moreover, we examined 4xHFS induced LTP between WT and cKO mice as the reviewer suggested. Similar as the 1xHFS induction, the LTP magnitude was also attenuated in cKO hippocampal slices even 2 hrs after LTP induction, indicating a long-term impairment of LTP in the cKO mice as well. The new result has now been incorporated into Figure 5H together with corresponding descriptions in the main text (line 208-210).

5. Hippocampal re-expression of KIF2C rescues mEPSC, LTP, and behavioral phenotypes. These are nice sets of data suggesting KIF2C directly regulates the synaptic and behavioral phenotypes. However, I wonder whether the same KIF2C re-expression rescues baseline and BDNF-dependent MT invasion, which is an essential part of the conclusions together with synaptic phenotypes.

We thank the reviewer for pointing this out. Following the suggestion, we over-expressed KIF2C in cKO neurons and examined the synaptic phenotypes carefully. We found that KIF2C re-expression was able to effectively rescue the baseline and BDNF-dependent MT invasion. The new result has now been incorporated into Figure 6E together with corresponding descriptions in the main text (line 243-247).

6. Biochemical work and ICC of hippocampal neurons suggests KIF2C is present postsynaptically, but it also colocalizes with Bassoon (Figure 1E). However, the authors discount the colocalization with a presynaptic label and only focus on the postsynaptic localization. A line scan is used to show colocalization, but this not sufficient. Colocalization analysis and Pearson's correlation coefficients should be computed to definitively show there is more colocalization with PSD95 than Bassoon.

We agree with the reviewer that KIF2C is localized both pre-synaptically and post-synaptically. The line scan result showed that KIF2C was colocalized with both PSD-95 and Bassoon. KIF2C must have played some roles in a pre-synaptic way. But in our electrophysiological experiments, we checked the paired-pulse ratios in hippocampus after KIF2C knockout and didn’t find any change (Figure 5B). Post-synaptic knockdown of KIF2C in CA1 neurons didn’t affect the paired-pulse ratios of CA3 neurons (Figure 3B). It is a hint that KIF2C may not be involved in the release probability. We didn’t perform the PDS-95 or Bassoon colocalization quantification because in our study we focused mainly on its function in post-synapse or dendrites. We found that the post-synaptic knockdown of KIF2C is sufficient to abolish the LTP in CA3-CA1 pathway (Figure 3F) and post-synaptic re-expression of KIF2C expression in CA1 could effectively rescue the LTP-defects in CA3-CA1 pathway (Figure 6G). Therefore, we think that at least KIF2C is an important post-synaptic regulator in neuron plasticity.

7. Throughout Figure 6 and Figure 7 the authors use a KIF2C point mutant (G491A) that has been shown previously to have little MT depolymerization activity. However, they do not directly compare the point mutant to the WT KIF2C when they attempt to rescue the synaptic transmission phenotypes in the cKO mouse. Instead they compare each to a control and use a Student's t-test to indicate that the cKO + WT protein is significantly different than control, while the cKO + G491A is not. They need to compare cKO + WT directly to G491A and controls via ANOVA and show these two "rescues" are different. Moreover, addition of a WT control would be needed if they wanted to claim that the cKO + WT is the same as WT control (i.e. a full rescue).

We thank the reviewer for the suggestion. In order to make our data more readable, we have compiled the data together and re-analyzed the result by one-way ANOVA. The results are consistent with previous ones and there was significant difference between WT and G491A rescue (Figure 6F-G, Figure 7F-H).

The re-expression of KIF2C in cKO mice was achieved *via* viral infection system, through which the expression level of KIF2C could be higher or lower than its endogenous level. Therefore, the phenotype readout of the “rescued mice” could be different to the WT mice. The aim of our rescue experiment was to prove the function of KIF2C in synaptic plasticity and cognition, and we did not claim “full rescue”. Moreover, according to our previous results, the expression level of KIF2C need to be finely tuned in neurons. Long-time overdosing of KIF2C affects synaptic plasticity or even cause deadly effect in neurons. In our study, we used doxycyline-induced expression to limit the duration and level of KIF2C expression. Even though, we did not speculate to reach a “full rescue” through viral-mediated overexpression but not the knockin approach. Therefore, we didn’t add the WT control in the rescue experiment.

8. In Figure 2 and 3 it is not clear why immunoblots are not used to show decreases in protein expression after knockdown. Only mRNA quantification is shown.

We have now added the immunoblot examination of KIF2C expression in Figure 2—figure supplement 1A, the result is consistent with the RT-PCR examination.

9. In Figure 2 KIF2C knockdown seems to have no effect on basal levels of spine density or spine head width. It is only after cLTP that significant differences are noted. However, in Figure 1D a Student's t-test is used, while in Figure 1E an ANOVA is used. Both of these graphs, as well as Figure 1G, should be analyzed with an ANOVA because more than two conditions are being compared. The minor differences (p<0.05) in Figure 1D and 1G are likely to be non-significant when analyzed with an ANOVA.

We suppose the panels the reviewer mentioned referred to those in Figure 2. We now have re-analyzed the data in Figure 2D by One-way ANOVA and the results are consistent. For Figure 2G, the bar chart we presented was misleading. What we actuary compared was the KIF2C expression level between the ECS and cLTP-treated groups in different fractions. We have now re-plotted the bar chart for clearer representation (Figure 2G).

10. Bar graphs and scatter plots are used randomly throughout the paper. Scatter plots, or better yet SuperPlots (Lord et al., 2020, JCB, 219(6):e202001064), should be used throughout the paper.

Thank you for pointing this out. We unified the chart format into scatter plot with bar graph in the revised manuscript.

11. The data in Figure 5E and F seem to be inconsistent. Both NMDAR and AMPAR appear to increase, although AMPAR to a greater extent than NMDARs, but the N/A ratio is significantly less in the cKO. Furthermore, only a p-value is provided. No statistical tests are mentioned.

We thank the reviewer for pointing this out. We think the immunoblotting and the electrophysiological approaches have different sensitivities and different experimental range so the results from the two experiments cannot be calculated and compared directly. To better address this question, we analyzed the input-output curve of AMPA/NMDA receptor-mediated EPSCs to better illustrate the changes of AMPARs and NMDARs in the patch clamp. KIF2C knockout alters the input-output curves of AMPA receptor but not that of NMDA receptor, indicating that KIF2C deficiency-caused EPSC amplitude alteration is AMPA receptor-mediated. The new result has now been incorporated into Figure 5D-E together with corresponding descriptions in the main text (line 200-202). We used Student’s *t-*test for statistical comparison.

12. In Figure 6B it is unclear why measurements of EB3 speed manually and using Tip Tracker software would result in such different growth speeds (EB3 speed vs. MT growth speed). The MT growth speeds are double the EB3 speeds. Shouldn't these be the same speeds?

We used EB3 to mark the +TIPs of MT and track the growing MT ends in live imaging. The EB3 speed we measured actually is the average velocity of EB3 moving (i.e., the displacement of EB3 divided by the time interval). As it cannot be measured by the software, we manually located EB3 at the beginning and at end of the live imaging and measured its displacement. The MT growth speed is the average forward moving speed of EB3. The software traced the moving of EB3 puncta and filters out the shrinking movement, calculates the average forward moving speed (i.e., total forward moving distances divided by the forward moving time). That’s why the MT growth speed is much higher than the EB3 speed. The detailed quantification procedures and parameters have been provided in the method section (Live-cell imaging and microtubule dynamics analysis, line 227-230, 510-527).

13. In Figure 7C (lines 260-261) the authors claim the maintenance of fear memory is impaired in KIF2C cKO mice, however it appears that the maintenance is preserved – the two lines are parallel. Rather the fear memory is decreased throughout the periods of measurement. Also, these measures need to be analyzed with a repeated measures ANOVA, rather than a t-test.

We thank the reviewer for the careful analysis. The freezing levels of WT and cKO mice did show similar decreasing tendencies. We re-checked the memory maintenance by calculating the percentage change of the freezing time between two consecutive times (i.e. (Freezing time_24h_- Freezing time_7d_)/ Freezing time_24h_ to contextual cue and (Freezing time_48h_- Freezing time_8d_)/ Freezing time_48h_ to conditioned stimulus) (Figure 7—figure supplement 1E) and there was no significant difference, showing the preserved maintenance of fear memory. We have corrected our mis-interpretation in the text (line 291-294).

Moreover, we have used RM ANOVA to analyze data in the revised submission, the result is consistent with our previous one.

14. Lines 320-21: The authors suggest that MTs bring AMPAR to the synapse. To my knowledge, this has not been shown here or elsewhere in the literature.

We thank the reviewer for pointing this out. We should draw our conclusions more carefully. We have now corrected the inappropriate description in the revised manuscript (line 355-359).

15. The model in Figure 8 is inaccurate. The authors want to show increased invasion of MTs into spines but show multiple simultaneous invasions. They never image MT dynamics in spines to show this.

We thank the reviewer for pointing this out. It is difficult to distinguish whether the increase in MT invasion frequency take place in the same spine or just the increase in invasion probability in a population of spines or both. By now, we cannot tell between the two possibilities, so we took down the schematic diagram.

16. The authors need to carefully review the writing contained within the manuscript, including the figure legends. The corresponding legend for Supplementary Information Figure S1 needs to be updated to reflect A-D as opposed to an abridged version A-B. Figure 4A includes "sgRNA" as opposed to "shRNA". Supplementary Information Figure S4 fails to mention B' in the legend and currently states "Number of strenching" as the title for the y-axis. There are also a number of confusing/distracting sentences within the body of the manuscript that need remedying.

We thank the reviewer for pointing this out. We have now revised the manuscript thoroughly and corrected the inappropriate wording or descriptions carefully.

17. In the description of Figure 2 (line 111), the authors state that spine morphology was examined at DIV 16-18 when the methods and figure legend state that this assessment occurred at DIV 18-20.

We are sorry for the mistake. We have now corrected the date.

18. Line 135: Figure S1 is not referenced regarding the cultured neurons.

We are sorry for the mistake. We had referenced Figure 2—figure supplement 1A but not the other panels in the text. We have now stated the other panels of Figure 2—figure supplement 1A in the corresponding part of the manuscript (line 120-123).

References:

1. F. J. Hoerndli et al., Kinesin-1 regulates synaptic strength by mediating the delivery, removal, and redistribution of AMPA receptors. Neuron 80, 1421-1437 (2013).

2. F. F. Heisler et al., GRIP1 interlinks N-cadherin and AMPA receptors at vesicles to promote combined cargo transport into dendrites. Proc Natl Acad Sci U S A 111, 5030-5035 (2014).

3. M. Setou et al., Glutamate-receptor-interacting protein GRIP1 directly steers kinesin to dendrites. Nature 417, 83-87 (2002).

4. C. C. Hoogenraad, A. D. Milstein, I. M. Ethell, M. Henkemeyer, M. Sheng, GRIP1 controls dendrite morphogenesis by regulating EphB receptor trafficking. Nat Neurosci 8, 906-915 (2005).

5. M. I. Monteiro et al., The kinesin-3 family motor KLP-4 regulates anterograde trafficking of GLR-1 glutamate receptors in the ventral nerve cord of *Caenorhabditis elegans*. Mol Biol Cell 23, 3647-3662 (2012).

6. H. Shin et al., Association of the kinesin motor KIF1A with the multimodular protein liprin-alpha. J Biol Chem 278, 11393-11401 (2003).

7. C. Maas et al., Synaptic activation modifies microtubules underlying transport of postsynaptic cargo. Proc Natl Acad Sci U S A 106, 8731-8736 (2009).

8. Y. Gutierrez et al., KIF13A drives AMPA receptor synaptic delivery for long-term potentiation via endosomal remodeling. J Cell Biol 220, (2021).

9. S. Uchida et al., Learning-induced and stathmin-dependent changes in microtubule stability are critical for memory and disrupted in ageing. Nat Commun 5, 4389 (2014).

10. A. Malgaroli, R. W. Tsien, Glutamate-induced long-term potentiation of the frequency of miniature synaptic currents in cultured hippocampal neurons. Nature 357, 134-139 (1992).

11. E. Fifkova, A possible mechanism of morphometric changes in dendritic spines induced by stimulation. Cell Mol Neurobiol 5, 47-63 (1985).

12. A. Vlachos et al., Synaptopodin regulates plasticity of dendritic spines in hippocampal neurons. J Neurosci 29, 1017-1033 (2009).

13. D. A. Fortin et al., Long-term potentiation-dependent spine enlargement requires synaptic Ca^2+^-permeable AMPA receptors recruited by CaM-kinase I. J Neurosci 30, 11565-11575 (2010).

[Editors' note: further revisions were suggested prior to acceptance, as described below.]

The manuscript has been improved but there are some remaining issues that need to be addressed, as outlined below:Specifically, the authors need to provide details on the statistical methods used through the manuscript and the reasons why the authors could not perform the suggested live imaging experiments (see below for details).Reviewer #3:The authors have taken the comments of the reviewers seriously and have responded with manuscript changes, additional experiments and moderated their interpretations of the data they present. These changes have strengthened the manuscript considerably. However, there are still a few points that the authors did not directly address.In their response to Essential Revision #1 the authors quantified the distribution of KIF2C with regard to PSD-95 in different ways but the data in Figure 2H and I are still not very convincing. They didn't seem to make any measurements in the dendrite shaft to show that KIF2C was increasing in the shaft while decreasing in the spine. From these data it seems they can just conclude that KIF2C is not colocalizing as much with PSD-95 after cLTP. Additionally, they state that they tried overexpressing KIF2C but this caused damage to the neurons. However, in Figure 6E they show they expressed GFP-KIF2C in cKO neurons. Why could they not image these neurons live and quantify changes in KIF2C in the spine and shaft as requested?The authors still used a Student t-test for Figure 2G and Figures 6F and G instead of ANOVA. Moreover, although they use ANOVA more appropriately in the revision, they have not indicated what post-tests they have run (Bonferroni, Tukey, etc.?) in their ANOVA analyses throughout the manuscript. These post-hoc tests should be included in the Methods and in the appropriate figure legends.

We thank the reviewer for pointing this out. we have added the quantitative data of KIF2C level in dendritic shafts after cLTP stimulation and clearly see an increase of KIF2C expression in dendrites accompanying a reduction in spines (Figure 2I), corresponding text was added to describe this finding (Line 133-138, 581-582).

We had tested KIF2C overexpression in WT and cKO neurons and slices with biochemical and electrophysiological experiments in our pre-tests, and finally found that neurons survive far less in wild type than in cKO, which was less than four days for cKO neurons in in vivo experiments and less than two dasys in cultured cKO neurons. For our rescue experiments in cKO neurons, overexpression of KIF2C(WT) was controlled in a limited time, or else a highly expressed KIF2C level would be toxic to cKO neurons. However, we found that the lowly expressed KIF2C in healthy cKO neurons showed very weaker fluorescent signal which was too dim to be detected under microscope. We therefore think it’s not suitable for the suggested live-imaging experiment. That’s why we used GFP antibody to detect KIF2C expression in the rescue experiment (Figure 6E).

For Figure 6F, G, we have re-analyzed the data by one-way ANOVA and the results are consistent with previous ones (line 1224 and 1234, Figure 6F, G). We have also carefully looked into our data, and added post-hoc tests in the Methods (line 814-821) and in the appropriate figure legends. For Figure 2G, what we actuary compared was the KIF2C expression level between the ECS and cLTP-treated groups in each fraction. We have now re-plotted the bar chart for clearer representation (Figure 2G).